# Molecular dynamics-based refinement and validation for sub-5 Å cryo-electron microscopy maps

**Abhishek Singharoy[1][*][†], Ivan Teo[1,2][†], Ryan McGreevy[1][†], John E Stone[1], Jianhua Zhao[3], Klaus Schulten[1,2][*]**

[1]Beckman Institute for Advanced Science and Technology, University of Illinois at Urbana-Champaign, Urbana, United States; [2]Department of Physics, University of Illinois at Urbana-Champaign, Urbana, United States; [3]Department of Biochemistry and Biophysics, University of California San Francisco School of Medicine, San Francisco, United States

**\*For correspondence:** singharo@ illinois.edu (AS); kschulte@ks.uiuc. edu (KS)

[†]These authors contributed equally to this work

**Competing interests:** The authors declare that no competing interests exist.

**Abstract** Two structure determination methods, based on the molecular dynamics flexible fitting (MDFF) paradigm, are presented that resolve sub-5 Å cryo-electron microscopy (EM) maps with either single structures or ensembles of such structures. The methods, denoted cascade MDFF and resolution exchange MDFF, sequentially re-refine a search model against a series of maps of progressively higher resolutions, which ends with the original experimental resolution. Application of sequential re-refinement enables MDFF to achieve a radius of convergence of ~25 Å demonstrated with the accurate modeling of β-galactosidase and TRPV1 proteins at 3.2 Å and 3.4 Å resolution, respectively. The MDFF refinements uniquely offer map-model validation and B-factor determination criteria based on the inherent dynamics of the macromolecules studied, captured by means of local root mean square fluctuations. The MDFF tools described are available to researchers through an easy-to-use and cost-effective cloud computing resource on Amazon Web Services.

## Introduction

Cryo-electron microscopy (cryo-EM) has evolved into one of the most effective structure determination tools in modern day structural biology, achieving in recent years resolutions rivalling those of X-ray crystallography or NMR spectroscopy (*Cheng, 2015*). Furthermore, cryo-EM based structure determination overcomes two major bottlenecks faced in traditional X-ray crystallography, namely, the arduous task of preparing well-ordered crystals of macromolecules (*Unger, 2002*), and the more fundamental problem with capturing these molecules in unphysiological states as a result of crystal contacts (*Neutze et al., 2015*). Consequently, cryo-EM provides a natural way of resolving the structures of large macromolecular complexes.

Historically, computational methods were required to bridge the resolution gap between crystallography and cryo-EM to produce atomic-resolution models of biomolecular complexes. Various real-space refinement methods that combine crystallographic structures and cryo-EM densities for structure determination have been developed, including DireX (*Schröder et al., 2007*), Flex-EM (*Topf et al., 2008*), Rosetta (*DiMaio et al., 2015*), FRODA (*Jolley et al., 2008*), Phenix real space refinement (*Afonine et al., 2013*), and Molecular Dynamics Flexible Fitting (MDFF) (*Trabuco et al., 2008*, *2009*; *McGreevy et al., 2016*).

MDFF, in particular, has proven to be an extremely successful refinement method as evidenced by its numerous applications (*Goh et al., 2015*; *McGreevy et al., 2016*) ranging from the intricate

**eLife digest** To understand the roles that proteins and other large molecules play inside cells, it is important to determine their structures. One of the techniques that researchers can use to do this is called cryo-electron microscopy (cryo-EM), which rapidly freezes molecules to fix them in position before imaging them in fine detail.

The cryo-EM images are like maps that show the approximate position of atoms. These images must then be processed in order to build a three-dimensional model of the protein that shows how its atoms are arranged relative to each other. One computational approach called Molecular Dynamics Flexible Fitting (MDFF) works by flexibly fitting possible atomic structures into cryo-EM maps. Although this approach works well with relatively undetailed (or 'low resolution') cryo-EM images, it struggles to handle the high-resolution cryo-EM maps now being generated.

Singharoy, Teo, McGreevy et al. have now developed two MDFF methods – called cascade MDFF and resolution exchange MDFF – that help to resolve atomic models of biological molecules from cryo-EM images. Each method can refine poorly guessed models into ones that are consistent with the high-resolution experimental images. The refinement is achieved by interpreting a range of images that starts with a 'fuzzy' image. The contrast of the image is then progressively improved until an image is produced that has a resolution that is good enough to almost distinguish individual atoms.

The method works because each cryo-EM image shows not just one, but a collection of atomic structures that the molecule can take on, with the fuzzier parts of the image representing the more flexible parts of the molecule. By taking into account this flexibility, the large-scale features of the protein structure can be determined first from the fuzzier images, and increasing the contrast of the images allows smaller-scale refinements to be made to the structure.

The MDFF tools have been designed to be easy to use and are available to researchers at low cost through cloud computing platforms. They can now be used to unravel the structure of many different proteins and protein complexes including those involved in photosynthesis, respiration and protein synthesis.

ribosomal machinery (*Villa et al., 2009*; *Trabuco et al., 2011*; *Frauenfeld et al., 2011*; *Wickles et al., 2014*) to a host of non-enveloped viruses (*Zhao et al., 2013*). So far this success has been limited to structure determination from typically low-resolution cryo-EM maps in the 7–25 Å range which, indeed, represented the state-of-the-art at the time of MDFF's inception (*Trabuco et al., 2008*). However, seminal advances in detection hardware and programs over the past three years (*Li et al., 2013*; *Milazzo et al., 2011*) have enabled now the routine availability of high-resolution (<5 Å) EM maps for a range of biological systems including ion channels (*Liao et al., 2013*), enzymes (*Bartesaghi et al., 2014*, *2015*), membrane fusion machinery (*Zhao et al., 2015*), and key functional components of the ribosome (*Fischer et al., 2015*; *Brown et al., 2015*).

High-resolution maps pose an imminent challenge to the traditional map-guided structure determination methods as the maps now characterize near-atomic scale features, the interpretation of which requires extremely precise structure building and validation protocols (*DiMaio et al., 2015*). For example, conformation of the protein sidechains, which are more flexible than the backbone, are now discernible within the maps and, thus, require precise modeling of the dihedral angles up to $C_\beta$ atoms while also respecting the map boundaries (*Barad et al., 2015*).

In order to produce atomic models with correct backbone and sidechain geometries, as well as minimal potential energy, structure determination tools must be augmented with chemically accurate force fields and exhaustive search algorithms respecting density constraints. Inspired by crystallographic modeling techniques, where such structure-building requirements have already been addressed for the resolution of 3–5 Å diffraction data (*DiMaio et al., 2013*; *Murshudov et al., 2011*; *McGreevy et al., 2014*), tools such as Rosetta have introduced Monte Carlo simulation-based segment building and refinement protocols with heuristic force fields (*DiMaio et al., 2015*), to handle high-resolution EM maps. Other notable automated model-building tools that can be used for

the refinement of high-resolution EM maps include Buccaneer (*Cowtan, 2006*), ARP/wARP (*Langer et al., 2008*), and Moulder (*Topf et al., 2006*).

Driven by a vision to extend the capabilities of flexible fitting approaches (*Topf et al., 2008*; *Trabuco et al., 2008*; *Tama et al., 2004*; *Suhre et al., 2006*; *Kovacs et al., 2008*; *Wu et al., 2013*) for addressing high-resolution maps, two new MDFF methods are introduced here. These methods, denoted cascade MDFF (cMDFF) and resolution exchange MDFF (ReMDFF), augment the traditional MDFF method (*Trabuco et al., 2008*, *2009*; *McGreevy et al., 2016*) (called direct MDFF henceforth) with enhanced conformational sampling techniques, namely simulated annealing (*Brünger, 1988*) and replica exchange molecular dynamics (*Sugita and Okamoto, 1999*). The central idea behind the techniques introduced is to fit a search model sequentially to a series of maps of progressively higher resolutions, ending with the original experimental resolution; all but the last in the series are computationally blurred lower-resolution derivatives of the original map, so that larger-scale features of the structure are determined first by fitting to the blurred densities, and smaller-scale refinements are performed subsequently during the fitting to higher-resolution densities. Altogether, this treatment enables a richer conformational sampling of the model within the map than direct MDFF, thereby allowing accurate modeling of the global and local structural features from the map; a similar treatment has previously been employed to increase the radius of convergence of MDFF protocols, but with crystallographic data (*Singharoy et al., 2015*).

The cMDFF and ReMDFF methods are demonstrated for structure analysis based on 3.2-Å and 3.4-Å resolution maps of β-galactosidase (*Bartesaghi et al., 2014*) and the TRPV1 channel (*Liao et al., 2013*), respectively. The two methods were found to resolve atomic structures with accuracy greater than that of direct MDFF and comparable to that of Rosetta, even with poor choices of search models. The accuracy is evaluated in terms of the quality of fit measured through global and local cross-correlations (GCC and LCC), integrated Fourier shell coefficients (iFSC), and EMRinger scores (*Barad et al., 2015*), as well as in terms of the quality of structural integrity measures like MolProbity (*Chen et al., 2010*).

In the second part of the present study we establish that structural flexibility, as measured by root mean square fluctuations or RMSF within the MDFF simulation, provides an ensemble-based indicator of local and overall resolution of a map offering, thus, a quality measure of an EM map based on the inherent dynamics of the imaged macromolecule. In line with this new finding, RMSF values are shown to provide a physical basis for the determination of optimal sharpening B-factors that maximize the signal-to-noise ratio within a map. These B-factors are determined at three different levels of model description: whole-system, per-domain, and per-residue.

Finally, use of the ReMDFF method on cloud computing platforms is discussed. Cloud computing is now a highly suitable approach for computational biology and can be employed for large-scale scientific computing, data analysis, and visualization tasks. For example, Amazon Web Services has been previously demonstrated to be a low-cost cloud computing platform for processing cryo-EM data (*Cianfrocco et al., 2015*). We demonstrate now the usage of Amazon Web Services, highlighting the platform's capability for rapidly fitting structures to EM density with ReMDFF. The web-interface makes it readily possible for experimental groups around the world to deploy MDFF in an easy and economical way, bypassing the need for their own staff, software, and hardware resources.

## Results

In the following section, we first describe the methodological advances achieved within cMDFF and ReMDFF for the resolution of sub-5 Å maps. Search model preparation, refinement, and structure validation protocols based on these advances are subsequently demonstrated for five exemplary protein complexes that were chosen based on the availability of high-resolution (3–5 Å) EM maps and atomic structures. Finally, the performance of ReMDFF on Amazon's cloud computing platforms is described, demonstrating that our MDFF software offers an efficient web-based resource for structure determination from EM maps.

### Simulation concept

In direct MDFF, an initial atomic structure is subjected to an MD simulation with an additional potential energy term $V_{EM}$ that is proportional to the sign-inverse of the EM map. Through $V_{EM}$, steering

forces locally guide atoms towards high-density regions, thereby fitting the structure to the map (see Materials and methods).

The equilibrium structure obtained in the simulation represents a global minimum in $V_{EM}$. For maps in the low resolution range (6–15 Å), this global minimum is broad, accomodating an ensemble of conformations defined by the overall shape of the macromolecule (*Trabuco et al., 2008, 2011*). In contrast, at the mid-resolution range of 4–6 Å, densities corresponding to the backbones become discernible, and at sub-4 Å resolutions, even sidechains can be resolved. At such high resolutions, $V_{\mathrm{EM}}$ now features multiple proximal local minima which correspond to recurring spatial patterns within a macromolecule, such as helices aligned in parallel or strands in a β-sheet. As shown in *Figure 1*, the energy barriers separating these local minima are typically twice as high as those in the case of low-resolution maps. The existence of such potential minima in high-resolution maps exposes MDFF to a long-known weakness of traditional MD-based algorithms, namely entrapment of the fitted structure within undesired local minima instead of reaching the global minimum of $V_{EM}$. Not unexpectedly, therefore, direct MDFF yields structurally poor or functionally irrelevant models with high-resolution EM maps (*Figure 2—figure supplement 1*) (*DiMaio et al., 2015*).

To mitigate this weakness in direct MDFF, we introduce cMDFF and ReMDFF. In cMDFF, the structure is fitted, in a series of MDFF simulations, to maps of gradually increasing resolution. First, the experimental map is smoothened by applying a series of Gaussian blurs with increasing halfwidths, $\sigma$, to obtain a set of theoretical maps with gradually decreasing resolution; $\sigma = 0$ Å corresponds to the experimental map, and $\sigma > 0$ Å corresponds to a smoothened one (see Materials and methods). Illustrated in *Figure 1*, the density-dependent potential derived from the smoothest map (i.e. the one with the largest $\sigma$ value) features a clear global minimum representing the large-scale structural features of the protein. Second, a search model is fitted to this map, allowing resolution of these large-scale features. Third, the resulting structure is employed as the search model for fitting to the next higher-resolution map in the series. These fitting and search model refreshment steps are repeated through the series of maps in order of decreasing $\sigma$, until the structure is finally fitted to the experimental map (see *Video 1* for TRPV1 refinement. An additional virtual reality version can be found at https://www.youtube.com/watch?v=UwwVC6C9tw0).

The gradual increase in map resolution over the course of the simulations allows the structure to explore a greater conformational space than in direct MDFF. The structure thus avoids entrapment within local minima of the density-dependent potential and is accurately fitted to the near-atomic density features of the experimental map while also resolving the larger scale features.

In ReMDFF, the cascade scheme is infused with a greater degree of automation. Multiple MDFF simulation replica are run in parallel with each replica fitting a model to a map of a specific resolution. Based on a Metropolis formula analogous to that of conventional replica exchange molecular dynamics simulations (*Sugita and Okamoto, 1999*), but now derived in terms of density and not temperature (see Materials and methods), the models are exchanged at regular time intervals between maps from neighboring pairs of replica. Stepwise improvements in fit occur during exchanges between a poorly fitted model at a high resolution with a well-fitted model at a lower resolution. This well-fitted model is further refined against the high-resolution map until convergence is reached, and exchange between the chosen resolutions ceases. Further details are described in Materials and methods.

ReMDFF has advantages over cMDFF both in terms of efficiency and automation as it can take advantage of modern parallel computing hardware and the powerful and adaptive replica exchange interface of NAMD (*Jiang et al., 2014*). Nonetheless, as presented in the following, both cMDFF and ReMDFF outperform direct MDFF in quality and speed across a range of high-resolution examples.

## Search model preparation and refinement

In an initial proof-of-principle computation, cMDFF and ReMDFF were applied to fit a structure of carbon monoxide dehydrogenase to a 3-Å synthetic density map. The same techniques were subsequently applied to obtain refined structures of two more protein systems, namely TRPV1 (*Liao et al., 2013*) and β-galactosidase (*Bartesaghi et al., 2014*), for which experimental densities of 3.4 Å and 3.2 Å resolution respectively are available. In each case, a direct MDFF simulation was also performed for the purpose of comparison. The MDFF-derived structures were then subjected to a model validation analysis, to evaluate the quality of the models with established protocols in the

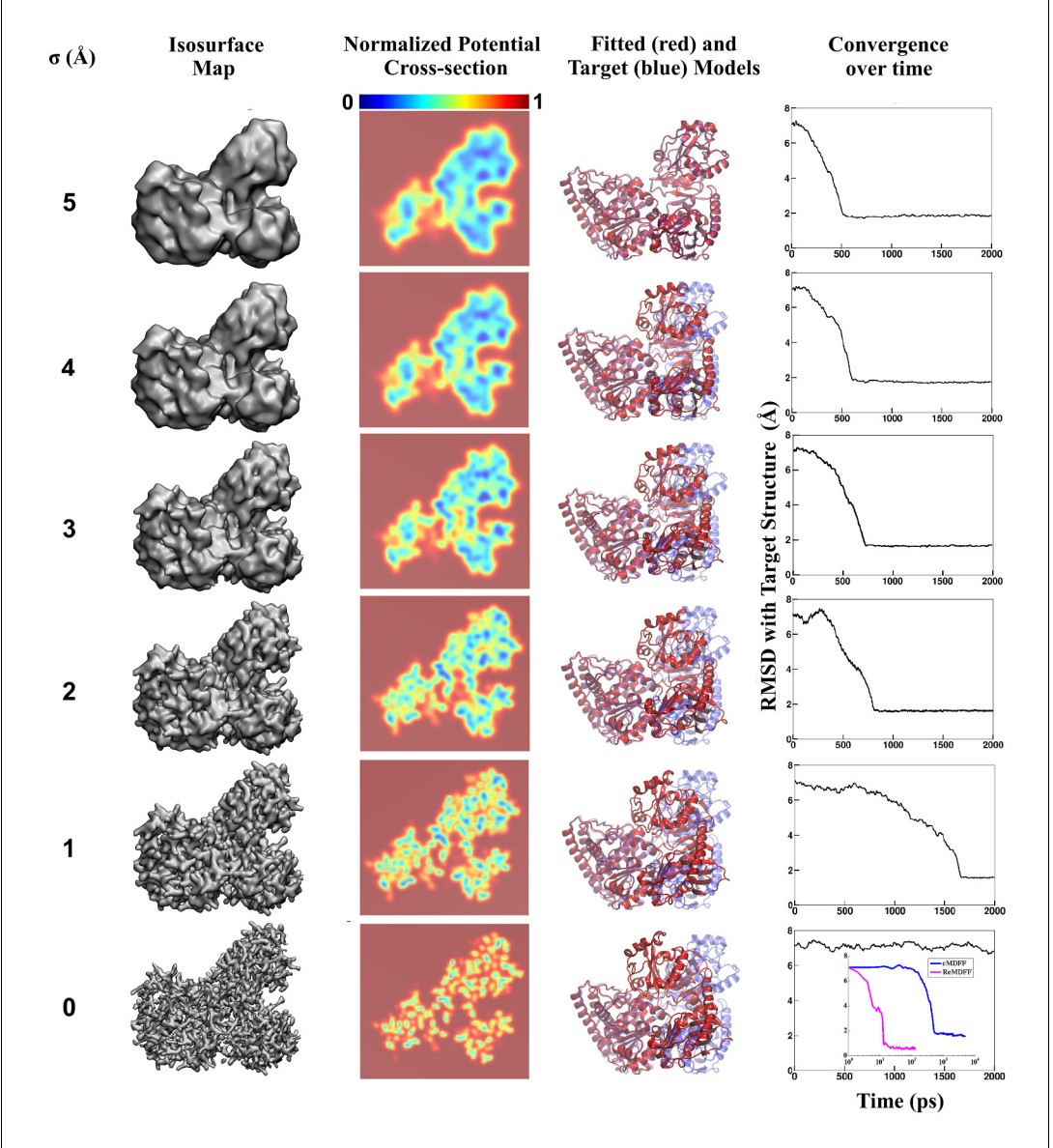

**Figure 1.** Visual summary of advanced MDFF methodology. A graphic table illustrating MDFF refinement of a model of carbon monoxide dehydrogenase using a high-resolution map. The map represents an open conformation while the initial search model was obtained through crystallography of a closed conformation. This search model was independently fitted, using direct MDFF, to individual members of a set of maps obtained by applying Gaussian blurs of various half-widths ($\sigma$, first column) to the experimental density. These maps are visualized as a 3D surface in the second column, while the resulting MDFF potentials $V_{EM}$ are represented in cross-section in the third column. Notice the increase in number of contiguous density regions as $\sigma$ increases. This increase in contiguity is manifested in the lowering of high $V_{EM}$ barriers (red) for small $\sigma$ values to low or flat energy profiles (blue) for larger $\sigma$ values, as observed in the $V_{EM}$ potential cross-sections. Reduced barrier heights allow the structure to explore the conformational space freely during fitting. The structure after 500 ps of fitting, shown in red, is superimposed on the known target structure, shown in blue, in the fourth column. The time evolution of RMSD with respect to the target during fitting is shown in the fifth column. The RMSD plots show that direct fitting to lower resolution maps requires fewer time steps to reach convergence. In fact, the structure never becomes less deviated than the initial 7-Å RMSD from the target in the direct MDFF of the highest-resolution map (i.e. in the absence of Gaussian blurring). The inset shows refinements of the same structure through cMDFF and ReMDFF employing the same set of maps. A clear improvement over direct MDFF is apparent, with convergence to within 1.7 Å and 1.0 Å of the target achieved within 1000 and 100 ps for cMDFF and ReMDFF respectively.

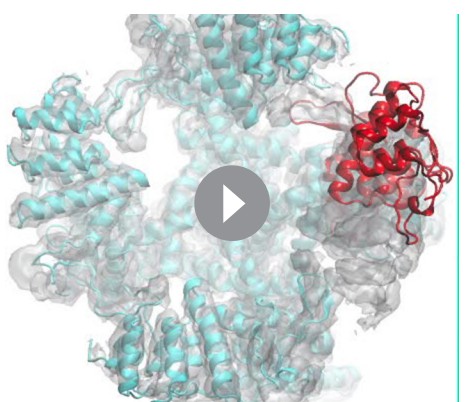

**Video 1.** cMDFF Refinement of TRPV1.

cryo-EM field. Additional examples chosen for this analysis include γ-secretase at 4.5 Å (*Lu et al., 2014*) and 3.4-Å (*Bai et al., 2015*), β-galactosidase at 2.2-Å (*Bartesaghi et al., 2015*) and the proteasome at 3.3 Å (*Li et al., 2013*) resolution. Comparisons between the direct and advanced MDFF protocols, and wherever possible, with other available fitting techniques, such as Rosetta (*DiMaio et al., 2015*), elucidate the general pros and cons of the flexible fitting strategy.

## Proof of principle

The performance of cMDFF and ReMDFF was evaluated on a test system, carbon monoxide dehydrogenase, which exhibits a closed and an open conformation (*Darnault et al., 2003*). Both these conformations have been crystallized, and are reported respectively in chains C and D of the PDB entry 1OAO. For our demonstration, the closed conformation (1OAO:chain C) was used as the search model, while the open one (1OAO:chain D) was the target.

First, a 3-Å resolution synthetic density map was constructed in Phenix (*Adams et al., 2010*), employing phases from the 1OAO structure and the associated diffraction data truncated at 3 Å. This map was then masked about chain D to yield a high-resolution envelope characterizing the open conformation. Assuming that the crystallographic model provides an accurate benchmark, the corresponding map for chain D determined here represents the best possible density data at 3 Å resolution that is experimentally attainable for the open conformation. Finally, through direct MDFF, cMDFF and ReMDFF, the search model constructed from chain C was fitted into this density to derive an atomic structure representing the open conformation. Both the cMDFF and ReMDFF refinements were performed for a set of six maps with $\sigma$ values ranging from 5 to 0 Å at constant decrements of 1 Å.

Accuracy of the fitting protocols was evaluated by comparing the fitted chain C structures with the crystallographically reported target chain D model. Direct MDFF of the 3 Å synthetic map performed for 2 ns converged to a structure with an RMSD of 7 Å relative to the target model. In sharp contrast, the cMDFF- and ReMDFF-generated structures are within 1.7 Å and 1 Å RMSD of the target (see the inset of *Figure 1*). It is also noted that fitting to the lowest-resolution (i.e. one with $\sigma$ = 5 Å) brings about an immediate decrease in RMSD from the target generating structures that are within 2 Å RMSD. Fitting of the structure to subsequent high-resolution maps brought the RMSD down to 1.0 Å.

The results demonstrate that the new protocols are capable of attaining well-fit structures where direct MDFF does not. In particular, one can think of the new protocols as extending the radius of

**Table 1.** β-galactosidase MDFF results. cMDFF and ReMDFF provide better fitted structures than direct MDFF according to various criteria. It is noteworthy that all structures refined by any form of MDFF display an improved MolProbity (*Chen et al., 2010*) score compared to the original de novo structure.

| Structure | RMSD(Å) | EMRinger | iFSC1(Å) | iFSC2(Å) | MolProb. | GCC |
|---|---|---|---|---|---|---|
| de novo (*Bartesaghi et al., 2014*) | 0.0 | 2.25 | 4.03 | 5.00 | 3.14 | 0.67 |
| Refined de novo | 0.6 | 4.23 | 4.19 | 5.20 | 1.23 | 0.68 |
| Initial | 7.7 | 0.24 | 0.14 | 0.15 | 1.49 | 0.48 |
| Direct MDFF | 3.7 | 2.31 | 2.11 | 2.74 | 1.38 | 0.56 |
| cMDFF | 0.7 | 3.16 | 4.22 | 5.22 | 1.37 | 0.67 |
| ReMDFF | 0.9 | 3.45 | 3.76 | 4.66 | 1.13 | 0.67 |

convergence to at least 7 Å, rendering the fitting procedures less dependent on the quality of the starting structure.

## Refinement of β-galactosidase

In a second test case, a search model was fitted into the 3.2 Å map (*Bartesaghi et al., 2014*) of β-galactosidase employing direct MDFF, cMDFF, and ReMDFF. Noting that the radius of convergence of the proposed MDFF protocols was at least 7 Å for the aforementioned test case, the initial search model was prepared such that it had an RMSD of 7 Å from the reported structure. This model was obtained by applying to the reported structure (obtained by de novo modeling within the EM map [*Bartesaghi et al., 2014*]) a high temperature MD protocol described in Appendix 1, Section 3 and, subsequently, choosing from the collection of trajectory structures one of RMSD 7 Å from the reported structure and with the lowest GCC with respect to the reported map (*Bartesaghi et al., 2014*) (*Figure 2—figure supplement 2a*).

Summarized in *Table 1*, the fitting results, in terms of quality of fit as well as model quality, are significantly better for cMDFF and ReMDFF than for direct MDFF: (**i**) RMSD of the fitted structure with respect to the reported de novo model is 0.7 Å and 0.9 Å for cMDFF and ReMDFF respectively, much lower than the 3.7 Å RMSD attained with direct MDFF (*Figure 2—figure supplement 3a*); (**ii**) EMRinger scores for cMDFF and ReMDFF are 3.16 and 3.45 respectively, higher than the 1.91 obtained for direct MDFF, implying accurate fitting of sidechains into the density; (**iii**) MolProbity scores are consistently small for all the flexible fitting techniques in part due to fewer, less severe steric clashes and fewer Ramachandran outliers (further detailed in *Table 2*); (**iv**) integrated FSC (iFSC2, corresponding to the range 3.4–10 Å on the FSC plot obtained as per Appendix 1 - Section 6), considered a more stringent measure of model quality than CC (*DiMaio et al., 2015*), attained higher values of 5.22 Å and 4.66 Å for cMDFF and ReMDFF, respectively, than 2.74 Å for direct MDFF. iFSC1, evaluated at the lower resolution range of 5–10 Å improves from 2.11 Å for direct MDFF to 4.22 Å and 3.76 Å for cMDFF and ReMDFF, respectively, showing a trend similar to that of iFSC2 corresponding to the high-resolution range; and (**v**) GCCs improved from an initial value of 0.48 to 0.56, 0.67 and 0.67 for direct, cMDFF, and ReMDFF protocols respectively. Similarly, typical residue LCC values improved from about 0 to greater than 0.80 (*Figure 2—figure supplement 4a* and *Figure 2—figure supplement 5*). Overall, cMDFF and ReMDFF refinements produce structures that interpret the 3.2-Å β-galactosidase map much more accurately than direct MDFF does.

**Table 2.** Structure quality indicators for β-galactosidase structures. β-galactosidase structures investigated in the present study were uploaded to the MolProbity server (http://molprobity.biochem.duke.edu) to extract the quantities presented below. The results show that the cMDFF- and ReMDFF-refined structures not only exhibit good measures of fit, but also improve the clash score and rotamer geometries, relative to the de novo and initial structures, while incurring only a small expense in Ramachandran statistics, bad angles, and Cβ deviations.

|  | de novo (*Bartesaghi et al., 2014*) | Refined de novo | Initial | Direct MDFF | cMDFF | ReMDFF |
|---|---|---|---|---|---|---|
| Clashscore | 53.7 | 0.0 | 0.0 | 0.0 | 0.0 | 0.0 |
| Poor rotamers (%) | 11.6 | 3.8 | 4.2 | 3.0 | 4.4 | 1.37 |
| Favored rotamers (%) | 67.4 | 90.8 | 87.8 | 92.1 | 89.8 | 95.3 |
| Ramachandran outliers (%) | 0.2 | 0.7 | 2.7 | 3.0 | 1.6 | 2.7 |
| Ramachandran favored (%) | 97.4 | 95.8 | 91.1 | 91.1 | 94.4 | 90.9 |
| MolProbity | 3.14 | 1.23 | 1.49 | 1.38 | 1.37 | 1.13 |
| Cβ deviations (%) | 0.0 | 0.05 | 4.92 | 0.18 | 0.29 | 0.39 |
| Bad bonds (%) | 0.09 | 0.04 | 3.61 | 0.02 | 0.01 | 0.03 |
| Bad angles (%) | 0.03 | 0.60 | 3.98 | 0.63 | 0.49 | 0.37 |
| RMS distance (Å) | 0.007 (0.025%) | 0.019 (0%) | 0.035 (0.237%) | 0.022 (0%) | 0.019 (0%) | 0.021 (0%) |
| RMS angle (degrees) | 1.1 (0.009%) | 2.2 (0.009%) | 3.6 (1.177%) | 2.4 (0.103%) | 2.1 (0.018%) | 2.3 (0.085%) |
| Cis prolines (%) | 8.06 | 8.06 | 6.45 | 6.45 | 6.45 | 8.06 |
| Cis non-prolines (%) | 1.15 | 1.15 | 0.0 | 0.0 | 1.15 | 0.0 |

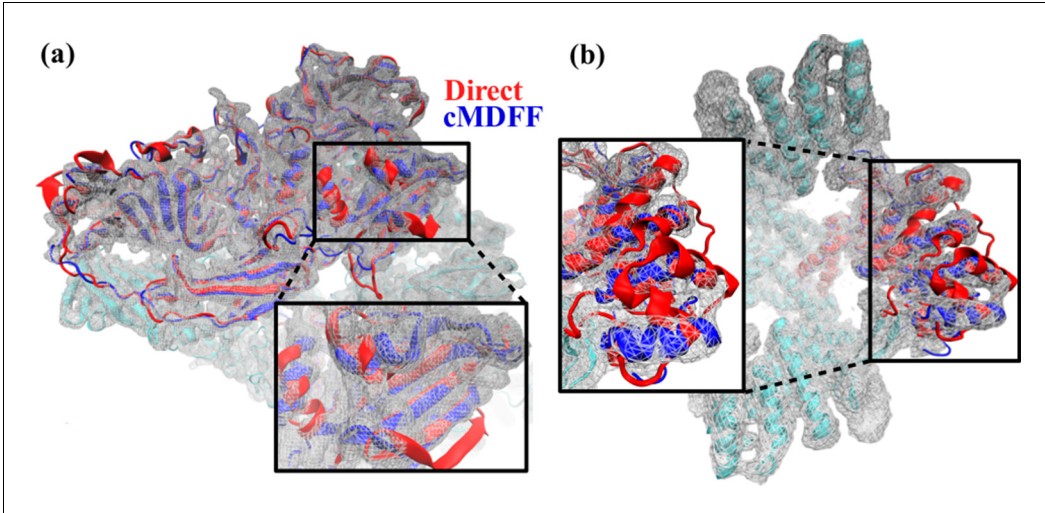

**Figure 2.** Comparison between cMDFF and direct MDFF fitted models. Models of (**a**) β-galactosidase and (**b**) TRPV1, obtained from cMDFF (blue) and direct MDFF (red) fitting simulations are superimposed. The cMDFF-fitted models fit well into the high-resolution maps (grey) of each molecule, whereas the direct MDFF models have become trapped in local minima that result in portions of the models protruding from the maps. ReMDFF-fitted models are almost identical to those from cMDFF and are therefore not shown.

The following figure supplements are available for figure 2:

**Figure supplement 1.** Global cross-correlation as a measure of fit.

**Figure supplement 2.** Comparison of initial models to target (published) models.

**Figure supplement 3.** Convergence of cMDFF, ReMDFF, and direct MDFF simulations.

**Figure supplement 4.** Local cross-correlations during cMDFF.

**Figure supplement 5.** Local cross-correlations during direct MDFF to refine de novo structures.

**Figure supplement 6.** Equilibration of cMDFF-refined model of β-galactosidase.

**Figure supplement 7.** Residues of β-galactosidase fitted within density map.

**Figure supplement 8.** FSC cross-validation plots.

*Figure 2a* shows, visually, how the cMDFF-derived structure differs from the direct MDFF structure in terms of fit. In judging the RMSD values to the target model the reader is reminded that equilibrium MD simulations of a single structure at room temperature typically exhibit RMSD values relative to the initial structure or the average structure of about 3 Å; the same is true for β-galactosidase (see *Figure 2—figure supplement 6* in Appendix 1 - Section 7). Consequently, an RMSD of 0.7 Å of the cMDFF/ReMDFF-fitted model relative to the target implies a high-quality refinement. The high quality of this refinement is further supported by visualizations of accurate sidechain placements within the density, shown in *Figure 2—figure supplement 7*.

The cMDFF- and ReMDFF-refined structures were found to be comparable in every quality measure in comparison to the reported de novo structure (*Bartesaghi et al., 2014*); in fact, the overall Molprobity and EMRinger scores are significantly better in cMDFF and ReMDFF. However, a closer look at the Molprobity score (*Table 2*) reveals that even though cMDFF vastly improves clash score and poor rotamers, it marginally increases the percentage of Ramachandran outliers and Cβ deviations relative to the de novo structure. Nonetheless, both cMDFF and ReMDFF improved structural

statistics with respect to the initial model (*Table 1*, third row) which was intentionally chosen to have a large deviation (RMSD of 7.7 Å) from the de novo structure.

Noting that the quality of MDFF output depends strongly on that of the search model, a second cMDFF simulation was also performed to refine the de novo structure within the reported map. The simulation, labeled 'refined de novo' in *Table 1*, yielded a structure that was superior in all the quality measures considered in comparison to the de novo structure as well as to the structures obtained from the various MDFF fittings of the other, 7.7 Å deviated initial model. A closer look at the Molprobity scores (*Table 2*) now reveals that not only are clash score and poor rotamers vastly improved, but the Ramachandran outliers and $C_\beta$ deviations are also comparable to the de novo structure.

A third cMDFF refinement was further performed with a search model of even lower structural quality (*Supplementary file 1E*, third row) compared to that of *Table 1* (third row). Structural statistics for the refinements from this lower-quality search model are provided in *Supplementary file 1F*. Comparison of the three cMDFF refinements starting with three search models of varying structural quality reveal that the poorer the secondary structure of the search model, measured in terms of higher percentages of rotamer and Ramachandran outliers, the worse the local structural statistics of the MDFF/cMDFF/ReMDFF-refined model. Surprisingly, large scale map-model validation measures, such as RMSD, GCC, or iFSC values, remain insensitive to such local discrepancies in the refined model. This insensitivity is apparent in the similarity of RMSD, GCC, and iFSC values of the three refined models of β-galactosidase (*Table 1* and *Supplementary file 1E*), which indeed feature very different number of Ramachandran outliers: 0.7% (*Table 2*-Refined de novo), 1.6% (*Table 2*-cMDFF) and 7.8% (*Supplementary file 1F*-cMDFF).

The high-precision of sub-5 Å maps demands correct assignment of secondary structure, which determines backbone geometry directly and sidechain conformations indirectly. In light of the initial model quality dependence of MDFF, it is advisable to begin refinement with an initial model of maximal secondary structure information, yet with minimal tertiary structure. Dependable initial secondary structure is also required since MD simulations have limited capability of recovering the structure of a protein fold if the search model begins with a random coil conformation (*Freddolino et al., 2010*). To this end, notwithstanding observed cMDFF improvements of the secondary structure (*Table 2* columns 4 vs. 6, and also demonstrated for γ-secretase [*Supplementary file 1D*]), moderate- to high-confidence homology models will serve as the most optimal starting point, as has been successfully shown for MDFF with low-resolution EM (*Chan et al., 2011*; *Noble et al., 2013*; *Wickles et al., 2014*) and crystallographic data (*McGreevy et al., 2014*; *Li et al., 2014*). Quantitatively speaking, employing the current example (β-galactosidase) and the following one (TRPV1), it is now shown that a search model which deviates from the target by an RMSD of 7–25 Å featuring Ramachandran outliers of ~3%, poor rotamers of 11–38%, overall Molprobity score of ~4 and EMRinger score of ~0.20 can be accurately refined against a sub-5 Å EM map with cMDFF or ReMDFF.

In terms of efficiency, the ReMDFF protocol exhibits the quickest convergence, arriving at steady state within 0.1 ns of simulation, whereas cMDFF requires around 0.8 ns. Both methods employed eleven maps with Gaussian blurs starting from a width of 5 Å and decreasing in steps of 0.5 Å towards the original reported map. To ensure that the cMDFF procedure did not over-fit the structures, cross-validation using EMRinger and FSC analysis was performed using half-maps from the EMD-5995 entry. iFSC and EMRinger values were found to be almost identical in both direct and cross comparisons. Details are provided in Appendix 1 - Section 11.

In addition to the MDFF simulations described so far for β-galactosidase, other simulations were performed to investigate in more detail the capabilities of MDFF. These simulations are described in Appendix 1 - Section 7. First, it was found that fitting of the $C_\beta$ atoms to the density is crucial for accurate placement of the sidechains. In agreement with prior EMRringer results (*Barad et al., 2015*), it is confirmed that MDFF placement of the backbone does not guarantee correct sidechain geometries, even with state-of-the-art CHARMM36 (*Klauda et al., 2010*) force fields. Second, MD simulation of the cMDFF-fitted β-galactosidase model revealed that this model is indeed an excellent representation of the most probable structures of the thermodynamic ensemble that characterizes the 3.2 Å map.

## Refinement of TRPV1

In the third test case, cMDFF and ReMDFF protocols were employed to fit an initial model to the 3.4-Å map (*Liao et al., 2013*) of the temperature-sensing protein TRPV1. The search model was prepared from the reported de novo structure through an interactive MD protocol described in Appendix 1 - Section 3. The model deviated from the de novo structure by an overall RMSD of 10 Å and locally, by about 25 Å in the vicinity of the ankyrin repeats represented by residues 199 to 430. This degree of deviation is in the ballpark of the lowest resolutions of usable EM maps and, therefore, represents the upper limit of uncertainty between a search model and the fitted structure that MDFF can still reconcile. Having to address an RMSD of 25 Å between search and target models, the present example represents an extreme test case for judging the radius of convergence of the proposed MDFF methods.

Fitting results for TRPV1, described in Appendix 1 - Section 8, were significantly better for cMDFF and ReMDFF than for direct MDFF, but now with much poorer search models than those employed for the β-galactosidase refinements. For example, cMDFF and ReMDFF refinements produced structures that interpret the 3.4-Å TRPV1 map within an RMSD of 2.4 Å and 2.5 Å from the target de novo model, much more accurately than does direct MDFF which converges to structures at an RMSD of 7.9 Å.

The cMDFF- and ReMDFF-obtained structures were found to be better in every overall quality metric in comparison to the de novo structure. *Figure 2b* illustrates the contrast in fit between the cMDFF and direct MDFF-derived structures. To observe the effect of MDFF on a substantially well-fitted initial structure, a direct MDFF simulation was also performed to refine the de novo structure within the reported map. The simulation yielded a structure that was comparable in all the quality measures considered to the structures obtained from cMDFF and ReMDFF (Appendix 1 - Section 8). However, the TRPV1 fitting results show that cMDFF and ReMDFF can have a radius of convergence as high as 25 Å in RMSD, whereas direct MDFF requires at the outset a well-fitted structure to deliver a satisfactory model. Also, since the number of Ramachandran outliers were minimal in both the 25 Å-deviated and de novo initial models, the cMDFF, ReMDFF, and direct MDFF-refined models exhibited low percentages of Ramachandran outliers, as reflected in *Supplementary file 1B*.

As was observed already for β-galactosidase, the ReMDFF protocol exhibited the quickest convergence, arriving at steady state within 0.02 ns of simulation, whereas cMDFF required around 0.27 ns. Both methods employed six maps with Gaussian blurs starting from a width of $\sigma$ = 5 Å and decreasing in steps of 1 Å to the reported $\sigma$ = 0 Å map. Cross-validation with half-maps was also performed on the cMDFF structure, as per the β-galactosidase simulations, to ensure that it was not over-fitted. As in the case of β-galactosidase, iFSC and EMRinger scores for direct and cross comparisons were similar. FSC analysis results are described in Appendix 1 - Section 11.

A separate set of model validation analyses was performed on the well-resolved TM portion of TRPV1 to pursue a direct comparison of a MDFF refined model with one from Rosetta (*Barad et al., 2015*). As reported in Appendix 1 - Section 9, MDFF produced results comparable to those of Rosetta when all the heavy atoms are coupled to the density: though the EMRinger score is marginally lower relative to that from Rosetta, the GCC and iFSCs are higher for MDFF; also MDFF provides a marginally higher MolProbity score. Altogether, major discrepancies between Rosetta and direct MDFF that were reported for the high-resolution EM maps (*DiMaio et al., 2015*) are now absent when employing the cMDFF and ReMDFF protocols, even with poorer choices of search models than those used with Rosetta. Thus, cMDFF and ReMDFF enable flexible fitting techniques to pursue resolution of structures within state-of-the-art maps obtained via cryo-EM.

## Model validation

An EM density map represents a thermodynamic ensemble of atomic conformations (*Schröder et al., 2007*; *Brunger et al., 2012*; *Schröder et al., 2010*). Conventionally, however, only a single model representing a best fit to the map is reported. One may ask how statistically representative a single model can be. To quantify the deviation of a fitted model from the rest of an ensemble of simulated molecules, root mean square fluctuation (RMSF) of the model relative to the ensemble-averaged structure was computed during an MDFF refinement simulation employing the protocol described in Methods. In the following, the RMSF of a fitted model is first shown to be indicative of the quality of fit of the model, as well as to represent the degree of natural

conformational variation exhibited within the thermodynamic ensemble underlying the map. Second, the RMSF values are found to correlate both locally and globally with the resolution of an EM map, providing an interpretation of map quality based on the inherent (i.e., natural) dynamics of the macromolecule under observation. Finally, RMSF values are also employed to identify optimal B-factor values for the sharpening of a map. Altogether, the results of the present study demonstrate that RMSF of a fitted model during an MDFF refinement provides valuable information on the model.

## RMSF and quality of fit

The relationship between RMSF values and quality of model fit is demonstrated for the cMDFF refinement of β-galactosidase at 3.2 Å resolution. The initial conformation is a poor fit of the map, characterized by low values of GCC, LCC, and iFSC (the row containing 'initial' structure in *Table 1*). Such conformations belong to a diverse ensemble of poorly fit structures, explored by the search model in the early phase of the fitting, that gives rise to high initial RMSF values shown in *Figure 4—figure supplement 1*. In the ending phase of the refinement, well-fitted structures are obtained with improved GCC, LCC, and iFSC (cMDFF row of *Table 1*). Owing to the high resolution (3.2 Å) of the EM density map, the population of these well-fitted structures is much smaller than that of partially-fitted structures. Thus, as can be observed in *Figure 4—figure supplement 1*, the converged ensemble of the well-fitted structures exhibit much smaller fluctuations than the initial one. Low RMSF values for the fitted structure indicate therefore, that (**i**) the structure has been modeled unambiguously within the map, and (**ii**) the structure can be regarded as representative of the ensemble underlying the 3.2-Å β-galactosidase map; since per-residue fluctuations about the fitted structure (*Figure 4—figure supplement 1*) are less than 1 Å, only marginal backbone and sidechain variations within the ensemble arise (*Singharoy et al., 2013*).

## RMSF and quality of map

Apart from representing the quality of fit, RMSF values monitored during an MDFF simulation correlate closely with the overall and local resolution of an EM map. Even though high-resolution cryo-EM data are becoming increasingly obtainable, resolution is not always uniform throughout a map. For example, *Figure 3* reflects the variation in local resolution of map regions corresponding to residues of β-galactosidase and TRPV1. Conformational flexibility can cause heterogeneity in the cryo-EM data (*Leschziner and Nogales, 2007*), producing local resolutions lower than that of the overall map.

Local resolution analysis (*Kucukelbir et al., 2014*) can be especially important for determining the parts of a high-resolution map that realistically contain side chain information and the parts that do not, preventing over-interpretation of the latter. MDFF protocols can be adjusted to account for such local variations and better inform the process of model validation. For example, if analysis shows that certain residues reside in low-resolution regions of the density, the per-atom weighting factor applied to the forces derived from the density can be lowered.

The RMSF-resolution correlation is found to hold in the cases of TRPV1 (PDB 3J5P), 2.2-Å β-galactosidase (PDB 5A1A), γ-secretase (PDB 5A63 and 4UPC), and the T20S Proteasome (PDB 3J9I). Generally, the lower the resolution of the map, the higher the corresponding overall RMSF during MDFF simulations. For example, the overall RMSF during MDFF of the 4.5-Å γ-secretase model and map is greater than that of the 3.4-Å model and map which, in turn, is greater than that of the 2.2-Å map and model of β-galactosidase (see RMSF labels on the upper column of *Figure 4*). The correlation between map resolution and model RMSF extends further to local features within the density. In *Figure 4* (upper row), RMSFs of atoms plotted against local resolutions of the corresponding map regions display linear correlation between the two quantities. Again, higher RMSF indicates lower local resolution.

The physical basis for considering RMSF values of a group of fitted atoms during an MDFF refinement as an indicator of the map resolution follows from the linear correlation of these values with RMSF of the same set of atoms during unbiased MD (*Figure 4*, bottom row). Noting that the RMSF value during unbiased MD simulations reflects flexibility (*Karplus, 1990*), this linear correlation clearly establishes the dependence of RMSF during MDFF refinement on the inherent flexibility of the macromolecule, at least for our four demonstration systems (see Appendix 1 - Section 2 for MD simulation details). Since the flexibility of a molecule during the imaging process contributes to the

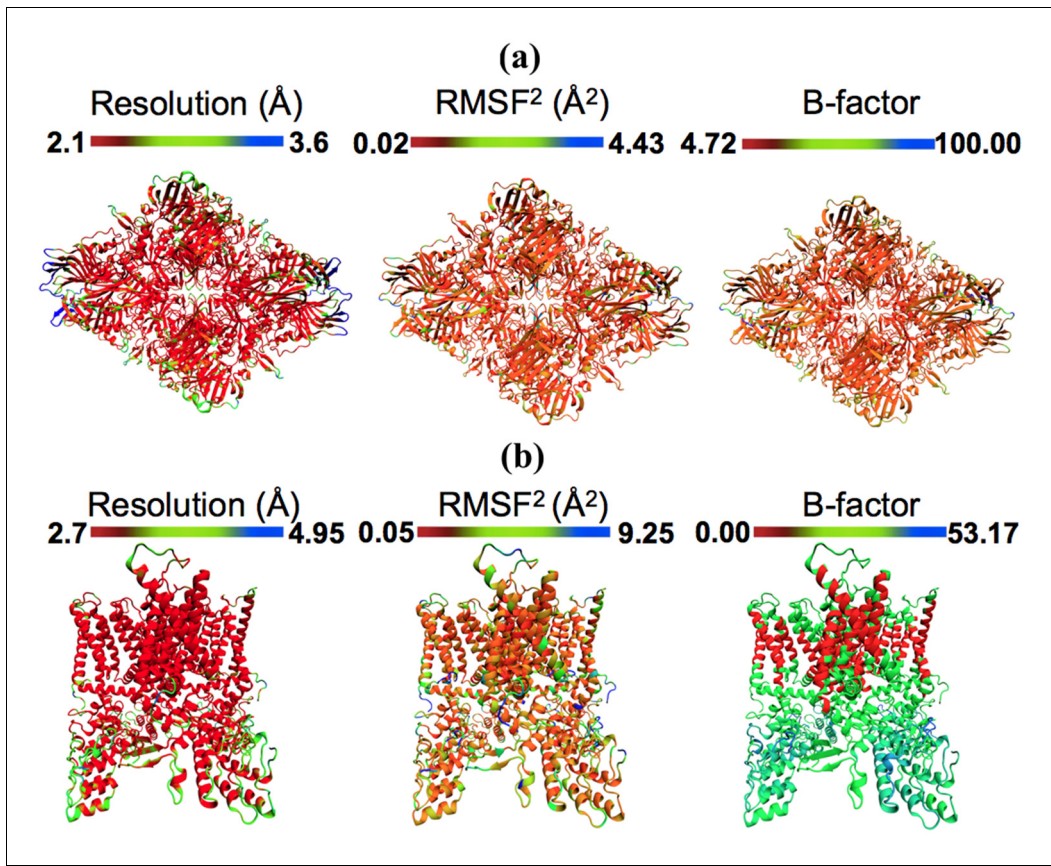

**Figure 3.** Models colored by local resolution, square of RMSF, and B-factor. The published models of (a) β-galactosidase (PDB 5A1A) and (b) TRPV1 (PDB 3J5P) are colored by the local EM map resolutions, the per-residue mean square fluctuations (RMSF$^2$) during MDFF simulation, and published B-factors. Comparison of these figures shows qualitative agreement between local resolution, RMSF$^2$, and B-factor. In fact, the local resolutions and B-factors correlate linearly with RMSF$^2$ of a fitted model both in the presence as well as absence of the EM map (more details in *Figure 4*).

limiting resolution of the resulting EM density map (*Kucukelbir et al., 2014*), it is not surprising that overall and local map resolutions correlate with overall and local RMSF values of the best fitted model (*Figure 4*).

In broad terms, through the present study we establish that RMSF, together with GCC, LCC, EMRinger (*Barad et al., 2015*), and iFSC, provide a comprehensive set of criteria for evaluating model and map quality on both global and local levels. The added value of RMSF is particularly evident on the local level, where the other measures may not perform as consistently. For example, a high LCC may be the result of a highly flexible structure fitting to a low-resolution region of the map, and not necessarily of a good representation of the local structure. As a result, although multiple low-resolution regions of the model in *Figure 4—figure supplement 2a* possess similar LCCs, disparate RMSFs of the same regions clearly indicate differences in local quality of the model. Likewise, EMRinger scoring, when applied to small groups of residues, does not correlate with local resolution (*Figure 4—figure supplement 2b*), and, therefore, is incapable of distinguishing regions of small number of atoms by local model quality. In contrast, RMSF clearly resolves local resolution, and, thus, resolves the map and model quality even with as few as 100 atoms.

## RMSF and B-factor determination
High contrast within an EM map allows clear identification of secondary structural elements. However, experimental imaging discrepancies arising from specimen movement and charging, radiation damage, and partial microscope coherence, or computational discrepancies due to inaccurate

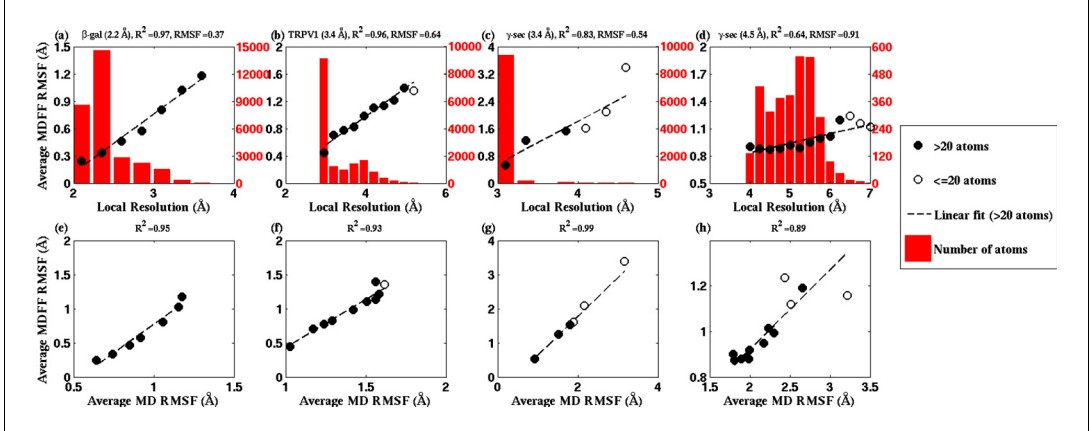

**Figure 4.** RMSF vs. local resolution plots for various simulations. For each test case shown, atoms in the MDFF-refined structure are classified by local resolution of the map regions they are fitted into. The average RMSF value of atoms (during MDFF simulation) in each resolution bin is calculated and plotted against the local resolution in the cases of (**a**) β-galactosidase (β-gal) at 2.2 Å, (**b**) TRPV1 at 3.4 Å, γ-secretase (γ-sec) at (**c**) 3.4 Å and (**d**) 4.5 Å resolution, and proteasome (see *Figure 4—figure supplement 3*). The numbers of atoms in the resolution bins are displayed as a histogram (in red) spanning a system-specific range of resolutions. The lowest resolution bins contained low (<20) populations and visual inspection consistently revealed the atoms to be on the edges of the density or were otherwise located inside map noise, and were therefore ignored during further analysis. A clear linear correlation between RMSF and local resolution can be found in each case, such that applying a linear fit produces the high $R^2$ value shown in each graph heading. Also displayed in each heading is an overall RMSF, averaged over all atoms in the system. The overall RMSF reflects the conformational variety of structures that fit within the map, and is found to correspond to the map resolution such that higher resolutions produce lower RMSFs. The second row of plots show that the RMSF during MDFF simulation also linearly correlates with RMSF during unbiased MD simulations of (**e**) β-gal, (**f**) TRPV1 and (**g,h**) γ-sec, establishing that fluctuations during MDFF reflect the inherent flexibility of a system.

The following figure supplements are available for figure 4:

**Figure supplement 1.** Per-residue RMSFs over β-galactosidase cMDFF fitting.

**Figure supplement 2.** EMRinger score and LCC do not predict local resolution in TRPV1.

**Figure supplement 3.** Average RMSF vs. local resolution during MDFF simulation of proteasome.

**Figure supplement 4.** RMSF values of individual residues during direct MDFF of published β-galactosidase models.

determination of the single particle order parameters (*Wade, 1992*; *Fernández et al., 2008*), introduce fuzziness to the EM map, thus hindering secondary structure identification. B-factor sharpening restores lost contrast by resolving the fuzzy features and, therefore, is a crucial step of map generation that affects map interpretation. Here, we describe the use of RMSF as a physical basis for the determination of optimal B-factors that preserve contrast during map sharpening.

*Figure 5*, *Figure 5—figure supplement 3*, *Figure 4—figure supplement 4* demonstrate the relationship between RMSF and B-factor for the 2.2-Å map of β-galactosidase, 3.4-Å map of TRPV1 and 4.5-Å map of γ-secretase; these three systems were chosen due to their reasonable size and availability of unsharpened data. An initial decrease in the RMSF values is observed with increase in B-factor sharpening of the map. However, the RMSF eventually reaches a minimum before increasing as the B-factor is further increased. Surprisingly, for all the three structures the B-factor corresponding to the minimum RMSF coincides with the one determined by Guinier analysis (*Fernández et al., 2008*; *Rosenthal and Henderson, 2003*) to provide maximum contrast. In fact, for TRPV1, B-factors determined from the RMSF minima are found to be higher for the soluble regions than for the transmembrane helices (*Figure 5—figure supplement 1*, b vs. c), again, in agreement with those B-factors derived from the Guinier analysis. Therefore, the RMSF analysis of maps with varying B-factor sharpening provides an alternate procedure for the determination of the optimal B-factor, which has been traditionally determined by Guinier analysis.

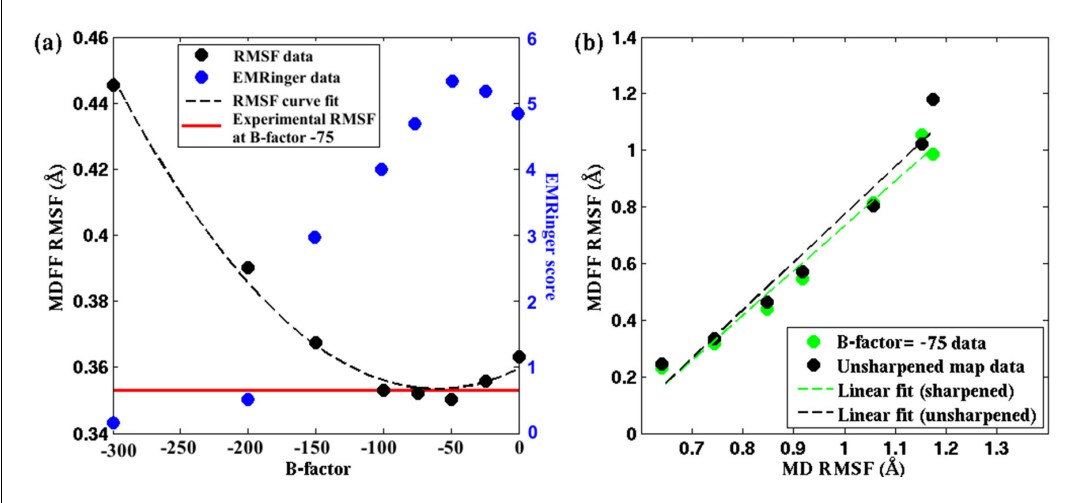

**Figure 5.** Effect of map sharpening on residue flexibility of β-galactosidase. (a) Overall RMSF of a fitted 2.2Å β-galactosidase structure (PDB 5A1A) during direct MDFF fitting as a function of the B-factor of the fitting map exhibits a parabolic trend. Guinier analysis identifies a B-factor of −75 as optimal, for which the corresponding RMSF (shown in red) coincides with the minimum of the trend line. EMRinger scores (shown in blue) of the same structures show a negative parabolic trend, with the peak coinciding with the minimum of the RMSF plot. (b) The linear relationships between local RMSF during MDFF and during unbiased MD for the unsharpened map and optimally sharpened map are compared. While the linear relationship is preserved in both cases, RMSFs in the sharpened case are slightly lower than in the unsharpened case.

The following figure supplements are available for figure 5:

**Figure supplement 1.** Effect of map sharpening on residue flexibility in TRPV1.

**Figure supplement 2.** Effect of map sharpening on residue flexibility in γ-secretase.

**Figure supplement 3.** EMRinger scores as a function of B-factor.

**Figure supplement 4.** Atom-by-atom B-factor for a β-galactosidase monomer.

The rationale for optimal B-factor selection based on an analysis of RMSF values is the following. As pointed out above and in *Figure 4*, RMSF of a fitted model is a function of the quality of the corresponding map. Following this argument, the maps in *Figure 5*, *Figure 5—figure supplement 1* and *Figure 5—figure supplement 2* that produce the model with the lowest RMSF during fitting represent the density envelopes where atoms can be positioned with the least uncertainty at the given experimental resolutions. The corresponding sharpening B-factor thus provides the optimal contrast for atom placement and secondary structure determination.

Our argument for the RMSF-based selection of B-factors is supported by the well-established quadratic relationship of B-factors with the RMSF of atoms of a structure in an experimental setting, i.e. $8\pi^2/3(RMSF)^2$ (*Rosenthal and Henderson, 2003*; *Liu and Xiong, 2014*). This relationship indeed implies that the smaller the average atomic fluctuations, the lower are the B-factors, and by definition (*Liu and Xiong, 2014*) the higher are the measured structure factor amplitudes, and hence the contrast, of the resulting map in both crystallographic and EM experiments (*Liu and Xiong, 2014*; *Fernández et al., 2008*). By analogy, RMSF within MDFF is a measure of these average atomic fluctuations under the experimental setting (*Figure 4*, bottom row) and, therefore, within a set of maps of varying B-factors, the highest contrast is indicated by the lowest RMSF obtained during fitting of a model. Since Guinier analysis of EM data selects B-factors based on the same atomic displacement-structure factor relationship that justifies the RMSF-based selection of B-factors (*Rosenthal and Henderson, 2003*), the B-factor selected from our analysis of the lowest RMSF matches excellently with those from the Guinier analysis of EM maps.

The higher contrast of the B-factor sharpened map, over the unsharpened one, is evident from the higher quality of the models derived from the sharpened maps. Presented in *Figure 5a*,

*Figure 5—figure supplement 3b*, and *Figure 5—figure supplement 1a*, for the cases of β-galacto-sidase, γ-secretase, and TRPV1, respectively, are EMRinger scores of models fitted to unsharpened and sharpened maps. These scores indicate clearly that the models derived from maps at B-factor sharpening of −50 (*Figure 5a*), −100 (*Figure 5—figure supplement 3b*), and −100 (*Figure 5—figure supplement 1a*) are more accurate than those derived from the unsharpened ones. Indeed, as further illustrated in *Figure 5a*, *Figure 5—figure supplement 3b*, and *Figure 5—figure supplement 1*, the maximum EMRinger score is attained for the same B-factor that produces the minimum RMSF. Sharpening by B-factors of magnitudes any higher than 50 for β-galactosidase, and 100 for γ-secretase or TRPV1 damages key density features thus reducing the quality of the associated fitted models. In addition, correlation of the RMSF of the fitted structure with that from an unbiased MD of the system is still preserved (*Figure 5b* and *Figure 5—figure supplement 2b*) for the B-factor sharpened map, confirming that the dynamical fluctuations of the structure within the sharpened map reflects the inherent dynamics of the system. Altogether, RMSF establishes a unique map-model validation criterion that represents an ensemble view of the fitted structures, while also preventing over-sharpening of the EM maps.

## RMSF and per-residue B-factors

The quadratic relationship between RMSF and B-factors can be further employed to determine per-residue B-factors. Presented in *Figure 3*, the per-residue $RMSF^2$ of β-galactosidase and TRPV1, derived from MDFF, show excellent agreement with the distribution of local resolution and experimentally reported B-factors: regions with higher $RMSF_2$ correspond to lower local resolution and higher per-residue B-factors, and vice versa. In fact, the quantitative agreement between the B-factors derived from MDFF through the computation of $8\pi^2/3(RMSF)^2$ ($0.02 < (RMSF)^2 < 4.43$ [*Figure 3*]) and those reported in the experiments is remarkable (*Figure 5—figure supplement 4*); a cross-correlation of 55% is found between the data sets, which improves to 60% in the structured regions. Larger discrepencies are observed between the RMSF-based and reported B-factors of TRPV1, particularly in the soluble region (*Figure 3b*). This discrepancy is expected, as for poorly resolved regions MD provides up to six-fold higher B-factors (*Kuzmanic et al., 2014*); the higher B-factors indeed have been demonstrated to be a more accurate representation of conformational diversity (*Kuzmanic et al., 2014*). For the TRPV1 example, the highest computed B-factor is $8\pi^2/3*9.25 = 240.5$ (*Figure 3b*) compared to the reported value 53.17. However, the majority of the computed B-factors fluctuatate about the value of $8\pi^2/3*1.73 = 44.98$, which is in fair agreement with the reported values of 20–30.

Overall, B-factors of cryo-EM maps are typically calculated by Guinier analysis for maps with resolutions better than ~10 Å (*Fernández et al., 2008*; *Rosenthal and Henderson, 2003*). Using a mask around specific regions of interest, estimation of local B-factors for different parts of a map is also possible. However, this method is limited to large domains of macromolecular complexes due to problems associated with tight masking of cryo-EM maps. For maps with resolutions better than ~3 Å, local B-factors could be estimated and refined in X-ray crystallography programs. However, most highresolution cryo-EM maps have resolutions between 3 to 5 Å (*Liao et al., 2013*; *Bartesaghi et al., 2014*; *Zhao et al., 2015*; *Fischer et al., 2015*). In this resolution range, B-factors could be estimated from MDFF and used as prior information to improve model building and refinement. For resolutions better than 3 Å, MDFF-derived values may serve as initial estimates of B-factors. Furthermore, many cryo-EM maps show local variations in resolution, complicating the model building process. A possible solution may be to combine local B-factor refinement for highresolution regions of a map and B-factors derived from MDFF for lower resolution regions. A resolution-dependent weighting scheme could be incorporated to combine the different values for optimal performance, which may help improve the accuracy of atomic models derived from high-resolution cryo-EM maps.

## Accessibility through cloud computing

ReMDFF involves many independent, though sporadically communicating, MDFF simulations that can be run well on a parallel computer as exploited by the NAMD software for the case of replica exchange simulations (*Jiang et al., 2014*). As a result, ReMDFF provides an efficient and automated method which can converge on a final fitted structure more quickly than direct or cMDFF (*Figure 2—*

**Table 3.** Performance and cost results for ReMDFF of carbon monoxide dehydrogenase on Amazon Web Services (AWS) Elastic Compute Cloud (EC2) platform. Costs are incurred on a per-hour basis, with a 1 hr minimum.

| Instance type | CPU | Performance (ns/day) | Time (hours) | Simulation cost ($) |
|---|---|---|---|---|
| c3.8xlarge | 30 | 5.88 | 0.41 | 1.68 |
| c3.4xlarge | 12 | 3.33 | 0.72 | 0.84 |
| c3.2xlarge | 6 | 1.35 | 1.78 | 0.84 |

*figure supplement 3*). However, a potential bottleneck may exist with respect to the computing hardware that a researcher has access to. Cloud computing offers a potential solution, allowing a researcher to focus on the scientific challenges of their project without having to worry about local availability and administration of suitable computer hardware.

To prove the feasibility of performing ReMDFF simulations on a cloud platform, we performed ReMDFF for a test system, carbon monoxide dehydrogenase (PDB 1OAO:chain C), on the Amazon Web Services (AWS) Elastic Compute Cloud (EC2) platform. The test system converges to the known target structure in approximately 0.1 ns simulation time. The time to convergence requires very little wall clock time, and, therefore, incurs a small monetary cost to a user (*Table 3*). However, it should be noted that human error and varying experience level can easily add to the incurred cost of cloud usage. Some systems may require multiple simulations to achieve a high quality structure and, therefore, additional time beyond the example discussed here. Furthermore, preparing a structure for simulation may require additional time and resources over the purely simulation-oriented results presented here. The files and information necessary to run ReMDFF on the test system using EC2 cloud computing resources are available at (http://www.ks.uiuc.edu/Research/cloud/). The Implementation section of the Methods contains further details for setting up and running ReMDFF simulations.

## Discussion

Flexible fitting methods have facilitated structure determination from low-resolution EM maps for more than a decade (*Tama et al., 2004*; *Suhre et al., 2006*; *Velazquez-Muriel et al., 2006*; *Orzechowski and Tama, 2008*; *Topf et al., 2008*; *Kovacs et al., 2008*; *Lopéz-Blanco and Chacón, 2013*; *Wu et al., 2013*) and continue to be the methods of choice for resolving molecular systems with atomic resolution. MDFF, in particular, has been a front-runner among methods that have facilitated the discovery of some of the most complicated structures in modern day structural biology (*Hsin et al., 2009*; *Sener et al., 2009*; *Gumbart et al., 2011*; *Frauenfeld et al., 2011*; *Zhao et al., 2013*; *Wickles et al., 2014*).

cMDFF and ReMDFF, the new variants of MDFF introduced above, offer now accurate fitting of atomic-level structures within sub-5 Å EM maps, a feat thus far inaccessible to direct MDFF. These new methods extend the radius of convergence of MDFF to at least 25 Å, fitting models to maps of resolutions as high as 3.2 Å. This radius of convergence is at least twice that reported for Rosetta refinements of the 20S proteasome (*DiMaio et al., 2015*). Such a broad radius of convergence will allow the refinement of extremely poorly guessed initial models with MDFF, as demonstrated in the cases of β-galactosidase and TRPV1 reported here.

ReMDFF simulations involving the so-called replica-exchange molecular dynamics converge quickly using a small number of replicas and are thus amenable to cloud computing applications. Running ReMDFF on the cloud lowers greatly the barrier to usage of the method, providing a cost-effective and practical solution to fitting structures to high-resolution cryo-EM densities for researchers who neither own nor can administer their own advanced computer hardware.

The accuracy of structures refined by cMDFF and ReMDFF has been confirmed by standard error analysis protocols, both in terms of quality of fit as well as in terms of the quality of the model. The results clearly show that sidechain refinements through MDFF produce accurate placement of $C_\alpha$ and $C_\beta$ atoms and modeling of the associated dihedrals.

Beyond the standard error analysis protocols, which apply to single static structures, the quality of both the fit and the model can be evaluated by ensemble-based measures. An example

demonstrated in this study pertains to the use of RMSF during MDFF refinement for simultaneously evaluating the quality of model, map, and fit. Furthermore, utilizing the fact that the inherent flexibility of a macromolecule is a key determinant of the achievable resolution of the corresponding map, RMSF values have been employed to identify the optimal amount of sharpening for a given map offering the highest contrast, as established typically through Guinier analysis. The RMSF computations in MDFF provide a viable means of determining per-residue B-factors.

Altogether, interpretation of a map as being representative of an ensemble rather than a single model brings to light new ways of model validation. EM maps, including those at high-resolution, typically do not have a uniform local resolution (*Kucukelbir et al., 2014*) and contain low-resolution regions, such as flexible exterior or transmembrane segments (*Leschziner and Nogales, 2007*). The sub-5 Å EM maps have also been able to resolve proteins in multiple conformations (*Matthies et al., 2016*). Both these type of maps will continue to benefit from accurate ensemble-based flexible fitting techniques for the foreseeable future. MDFF provides a natural method to model flexible regions; de novo models constructed for one conformational state can be flexibley fitted into the density of the other state(s), thus avoiding the arduous task of model construction for all the conformational states for capturing a conformational transition process with cryo-EM.

## Materials and methods

In the following section, we first outline the methodology underlying direct MDFF, cMDFF, and ReMDFF, along with specific protocols applied for the refinement of β-galactosidase and TRPV1 models. Second, we describe the computations of local and global root mean square fluctuation (RMSF) values, that are utilized for the evaluation of map-model quality. Finally, details on the computational implementation of all the three MDFF protocols are provided.

### Direct MDFF

MDFF requires, as input data, an initial structure and a cryo-EM density map. A potential map is generated from the density and subsequently used to bias a MD simulation of the initial structure. The structure is subject to the EM-derived potential while simultaneously undergoing structural dynamics as described by the MD force field.

Let the Coulomb potential associated with the EM map be $\Phi(\mathbf{r})$. Then the MDFF potential map is given by

$$V_{\mathrm{EM}}(\mathbf{r}) = \begin{cases} \zeta\left(\frac{\Phi(\mathbf{r}) - \Phi_{\mathrm{thr}}}{\Phi_{\mathrm{max}} - \Phi_{\mathrm{thr}}}\right) & \text{if } \Phi(\mathbf{r}) \geq \Phi_{\mathrm{thr}}\,, \\ \zeta & \text{if } \Phi(\mathbf{r}) < \Phi_{\mathrm{thr}}\,. \end{cases} \tag{1}$$

where $\zeta$ is a scaling factor that controls the strength of the coupling of atoms to the MDFF potential, $\Phi_{\mathrm{thr}}$ is a threshold for disregarding noise, and $\Phi_{\mathrm{max}} = \max(\Phi(\mathbf{r}))$. The potential energy contribution from the MDFF forces is then

$$U_{\mathrm{EM}} = \sum_i w_i V_{\mathrm{EM}}(\mathbf{r}_i)\,, \tag{2}$$

where $i$ labels the atoms in the structure and $w_i$ is an atom-dependent weight, usually the atomic mass.

During the simulation, the total potential acting on the system is given by

$$U_{\mathrm{total}} = U_{\mathrm{MD}} + U_{\mathrm{EM}} + U_{\mathrm{SS}} \tag{3}$$

where $U_{\mathrm{MD}}$ is the MD potential energy as provided by MD force fields (e.g. CHARMM) and $U_{\mathrm{SS}}$ is a secondary structure restraint potential that prevents warping of the secondary structure by the potentially strong forces due to $U_{\mathrm{EM}}$. A detailed description of the potentials arising in *Equation 3* is given in Trabuco et al (*Trabuco et al., 2008, 2009*).

After the MDFF and restraint potentials are created through the MDFF plugin of VMD (*Humphrey et al., 1996*), the initial structure is rigid-body docked (e.g. with Situs [*Wriggers, 2010*]) into the density map. Prior to simulation, MDFF-specific parameters can be modified and include $\zeta$ and the subset of atoms to be coupled to the MDFF potential. The latter typically consists of all non-

hydrogen atoms or backbone atoms and $\zeta$ is usually set to 0.3. MDFF can be performed in various simulated conditions, including different temperatures and vacuum, membrane, and explicit or implicit (*Tanner et al., 2011*) solvent environments. The choice of parameters and conditions depends on the requirements of each specific case. For example, a highly polar molecule would be more accurately simulated in explicit solvent rather than in vacuum, but the computational cost would be much higher in this case. The MDFF simulation is run until the system has reached stationarity, as determined by RMSD; typical run times are nanoseconds.

## Cascade MDFF

In cascade MDFF (cMDFF), the initial structure is sequentially fitted to a series of potential maps of successively higher resolution, with the final potential map being the original one derived from the EM map. Starting with $i = 1$, the $i$th map in the series is obtained by applying a Gaussian blur of width $\sigma_i \geq 0 \sim \text{Å}$ to the original potential map, such that $\sigma_i$ decreases as the structure is fitted in the sequence $i = 1, 2, \ldots, L$, where $L$ is the total number of maps in the series, so that $\sigma_L \geq 0 \sim \text{Å}$. One can intuitively understand cMDFF as fitting the simulated structure to an initially large and ergodic conformational space that is shrinking over the course of the simulation towards the highly corrugated space described by the original MDFF potential map.

*Figure 1* provides a visual representation of cMDFF. For a mathematical illustration, suppose that the original potential map can be written as a sum of Gaussians

$$V_{\text{EM}}(\mathbf{r}) = \sum_n c_n G\big(\mathbf{r}\,; \mathbf{r}'_n, s_n\big),$$ (4)

where $c_n$ is a weight, $G\big(\mathbf{r}\,; \mathbf{r}'_n, s_n\big)$ is a Gaussian function centered at $\mathbf{r}'_n$ with half-width $s_n$ and evaluated at $\mathbf{r}$. The result of a Gaussian blur of half-width $\sigma_i$ on $V_{EM}$ is (see Appendix 1 - Section 1 for details)

$$V_{\sigma_i}(\mathbf{r}) = \sum_n c_n G\left(\mathbf{r}\,; \mathbf{r}'_n, \sqrt{s_n^2 + \sigma^2}\right).$$ (5)

Hence, the half-width $\sigma_i$ allows one to tune the characteristic width of the potential map through the half-widths of its component Gaussians $\sqrt{s_n^2 + \sigma_i^2}$. The initial fitting starts with a large $\sigma_1$, corresponding to a diffuse potential which allows much structural mobility, and proceeds along decreasing values of $\sigma_i$, corresponding to narrower potentials with steeper gradients, so that the structure is gradually settled into the original potential map, characterized by $\sigma_L \geq 0 \sim \text{Å}$.

In practice, the series of cMDFF maps is generated from the original potential map using VMD's volutil Gaussian blur tool. Optimal values for the first half-width $\sigma_1$ and the change in $\sigma_i$ from one map to the next are case-dependent. Values used in the present study were obtained through trial and error. In general, structures far from the ideal conformation benefit from a large $\sigma_1$ (>5 Å) so as to explore a large conformation space. On the other hand, if the original map has a high resolution, small changes in $\sigma_i$ (<1 Å) would allow a gradual convergence required to avoid being trapped in local potential minima. In our simulations, the change in $\sigma_i$ is initially 1 Å. A concrete example is $\sigma_1 = 5$ Å, $\sigma_2 = 4$ Å, $\sigma_3 = 3$ Å, $\sigma_4 = 2$ Å, $\sigma_5 = 1$ Å, $\sigma_6 = 0$ Å. A second trial using changes of 0.5 Å was performed ($\sigma_1 = 5$ Å, $\sigma_2 = 4.5$ Å, $\sigma_3 = 4$ Å,..., $\sigma_{10} = 0.5$ Å, $\sigma_{11} = 0$ Å), and if the resulting structure of the second trial presented a better fit, then the first trial was discarded.

## Resolution exchange MDFF

Replica Exchange MD (ReMD) is an advanced sampling method that explores conformational phase space in search of conformational intermediates, which are separated by energy barriers too high to be overcome readily by fixed temperature simulations. Instead of working with a single, fixed MD simulation, ReMD carries out many simulations in parallel, but at different temperatures $T_1 < T_2 < T_3 < \ldots$ where the lowest temperature $T_1$ is the temperature of actual interest, typically, room temperature. The simulations of several copies of the system, the so-called replicas, run mainly independently, such that ReMD can be easily parallellized on a computer, but at regular time points the instantaneous conformations of replicas of neighboring temperatures are compared in terms of energy and transitions between replicas are permitted according to the so-called Metropolis

criterion (**Sugita and Okamoto, 1999**). This way the highest temperature replicas overcome the energy barriers between conformational intermediates and through the Metropolis criterion moves the $T_1$ replica benefits from it such that transitions between intermediates occur frequently. The application of the Metropolis criterion in the protocol guarantees that the conformations of the $T_1$ replica are Boltzmann-distributed.

ReMDFF extends the concept of ReMD to MDFF by simply differentiating replicas not by temperatures $T_1 < T_2 < T_3 < \ldots$, but by the half-width parameters $\sigma_1 > \sigma_2 > \sigma_3 > \ldots$. Numerical experiments showed that ReMDFF works extremely well as documented in the present study. As NAMD can parallelize ReMD well (**Jiang et al., 2014**), it can do the same for ReMDFF, such that the enhanced sampling achieved translates into extremely fast MDFF convergence. At certain time instances replicas $i$ and $j$, of coordinates $\mathbf{x}_i$ and $\mathbf{x}_j$ and fitting maps of blur widths $\sigma_i$ and $\sigma_j$, are compared energetically and exchanged with Metropolis acceptance probability

$$p(\mathbf{x}_i, \sigma_i, \mathbf{x}_j, \sigma_j) = \min\left(1, \exp\left(\frac{-E(\mathbf{x}_i, \sigma_j) - E(\mathbf{x}_j, \sigma_i) + E(\mathbf{x}_i, \sigma_i) + E(\mathbf{x}_j, \sigma_j)}{k_B T}\right)\right), \tag{6}$$

where $k_B$ is the Boltzmann constant, $E(\mathbf{x}, \sigma)$ is the instantaneous total energy of the configuration $\mathbf{x}$ within a fitting potential map of blur width $\sigma$.

## MDFF protocols for β-galactosidase and TRPV1

Computational protocols for performing direct MDFF, cMDFF, and ReMDFF refinements of the two test systems, β-galactosidase and TRPV1, are now outlined.

### Direct MDFF

In order to provide a basis for comparison with cMDFF and ReMDFF, direct MDFF simulations were performed for both β-galactosidase and TRPV1. Original published maps of resolutions 3.2 Å and 3.4 Å for β-galactosidase (EMD-5995 [**Bartesaghi et al., 2014**]) and TRPV1 (EMD-5778 [**Liao et al., 2013**]), respectively, were fitted with search models characterized by RMSD values of 7 Å and 25 Å, respectively, relative to the known de novo target models (PDB entries 3J7H and 3J5P, respectively); these search models were prepared as described in Appendix 1 - Section 3. Scale factors $\zeta$ (**Equation 1**) of values 1.0 and 0.3 were employed for β-galactosidase and TRPV1, respectively, to couple all the heavy atoms of the models to the respective maps. All other simulation parameters are noted in Appendix 1 - Section 2. The resulting structure from each MDFF simulation was then subjected to a final re-refinement applying a scaling factor of 1.0. Furthermore, the temperature was ramped down from 300 K to 0 K over 30 ps and held at 0 K for an additional 1 ns. This re-refinement step additionally improved the fitting of sidechains (**McGreevy et al., 2014**). Thus, all the MDFF results reported in the present study pertain to the structures that are obtained from direct MDFF, cMDFF, or ReMDFF simulations followed by the final re-refinement step.

### cMDFF

The general simulation protocol consists of a series of consecutive MDFF simulations. The search models for β-galactosidase and TRPV1 were first minimized over between 500 to 1000 time steps. Next, MDFF simulation runs were sequentially performed, starting with the map of the lowest resolution, progressing through maps of successively higher resolution, and ending with the original map. Each run was chosen to be long enough for the structure to equilibrate within the MDFF potential. Taking advantage of the stochastic nature of MDFF simulations, multiple independent cMDFF simulations (10 in the present study) were performed for each system to be fitted, generating an ensemble of fitted structures. From the ensemble, the best structure was determined by the various quality indicators described in Results. This structure was then subjected to the final re-refinement step to allow for accurate resolution of sidechains.

For β-galactosidase, cMDFF was initiated with a map blurred with half-width $\sigma_1 = 5$ Å. Use of 0.5 Å resolution steps produced better-fitted structures than 1 Å steps. Hence, the final simulation utilized $L = 11$ maps in total, including the original. A search model was obtained by subjecting the published structure to a 5-ns equilibration MD run with temperature 1000 K, yielding a structure with backbone RMSD of 7.6 Å relative to the original structure (see Appendix 1 - Section 3). 70-ps MDFF simulations were performed at each of the 11 resolutions to achieve convergence during the

cMDFF protocol. Simulation parameters for all these MDFF runs are identical to those used for the direct MDFF simulations above.

For TRPV1, Gaussian blurred maps were generated starting with a half-width of $\sigma_1 = 5$ Å, and decreasing by 1 Å for each subsequent map, thus yielding a series of $L = 6$ maps, including the original. The initial structure for this case was obtained using an interactive MD protocol, described in Appendix 1 - Section 2, to displace the ankyrin repeat region of one subunit. The resulting model deviated from the de novo model by an RMSD of about 10 Å in the displaced region and about 25 Å in the overall structure. Unlike β-galactosidase, now, 100-ps MDFF simulations were performed at each of the 6 resolutions to achieve convergence.

### ReMDFF

ReMDFF was performed on both β-galactosidase and TRPV1 using the same search models, simulation parameters, and high-resolution maps as in the cMDFF simulations. 11 and 6 replicas were employed for β-galactosidase and TRPV1 respectively with an exchange trial interval of 1 ps. In each case, the total energy of each replica was monitored and the simulation was run until the energies reached a stationary level. The ReMDFF simulation was found to converge in 0.1 ns for the β-galactosidase refinement, and in 0.02 ns for that of TRPV1. Finally, similar to direct MDFF and cMDFF, the re-refinement step was performed to improve sidechain geometry.

## Fluctuation analysis

The local resolutions of a density map can be computed with ResMap (*Kucukelbir et al., 2014*) and used within VMD to select the atoms of a structure that are contained in a range of resolutions found by the ResMap analysis. First, the local resolution map output by ResMap is loaded into VMD and then the *interpvol* keyword can be used to automatically select the atoms found inside the volume values specified, using interpolation. The average RMSF of each selection can then be calculated for a structure during the steps of a MDFF simulation after the initial fitting has occured and the structure has stabilized. In principle any criteria for atom selection can be used for RMSF analysis, though we use local resolution of the EM density here to illustrate the correlation between the two measurements. Additionally, we compute a global average RMSF of the entire structure.

The ensemble-based nature of the RMSF analysis means that the quality metric is not dependent on a single structure, but instead a large family of structures can be employed as a better representative of the data. Ensemble-based analyses are a natural and powerful benefit of the MD-based nature of MDFF. RMSF analysis does not, however, require MDFF to be used as the method of refinement. In principle, any refinement method can be used to obtain the fitted model. A subsequent short MDFF simulation of the fitted model can then be performed to obtain the data necessary for the RMSF analysis.

## Implementation

Incorporating advanced simulation techniques, such as multi-copy algorithms (*Jiang et al., 2014*), into the MDFF protocol creates a more efficient and accurate computational strategy in cMDFF and ReMDFF. However, these advanced simulation techniques come with an added complexity in the setup and execution of the methods. The current implementation of these methods in NAMD (*Phillips et al., 2005*) and VMD (*Humphrey et al., 1996*) aim to automate the steps previously discussed. The MDFF Graphical User Interface (GUI) (*McGreevy et al., 2016*) can be used to set up cMDFF and ReMDFF simulations and provides default parameters, including the number and extent of smoothed maps used for the fitting, with which to run. The parameters for the smoothed maps and number of steps used per map are set heuristically based on previous experience and represent an adequate initial starting point. The GUI automatically generates each of the smoothed maps and converts them to potentials for use in the ReMDFF simulation. All parameters can be tuned by a user to adapt the protocols to their specific system and preference. The GUI produces a series of NAMD configuration files and scripts used for running the simulation, as well sorting and visualizing the results in VMD. Instruction on the use of MDFF, including the GUI, is given in the tutorial found at http://www.ks.uiuc.edu/Training/Tutorials/science/mdff/tutorial_mdff-html/.

Future development will allow for the automatic generation of the smoothed maps in NAMD at runtime. NAMD will also analyse the dynamics of the system to determine when the simulation has

converged and move on to fitting to the next density map in the sequence in case of cMDFF calculations. Furthermore, advanced visualization and analysis techniques in VMD (e.g. new graphical representations) will be critical for properly understanding the RMSF analyses and to provide greater insight when examining the quality of a model.

The use of advanced simulation techniques also comes with an added cost of computational requirements. Adapting the algorithms to best utilize available computational hardware is key when developing efficient methods. Fortunately, the cMDFF and ReMDFF methods and associated analysis algorithms are well suited to highly efficient software implementations on contemporary multi-core CPUs and graphics processing unit (GPU) accelerators. We observe that by storing the complete cascade resolution series in efficient multi-resolution data structures such as mip maps (*Williams, 1983*), the MDFF cascade algorithm can access a continuously variable resolution representation of the original cryo-EM density map, while making efficient use of CPU and GPU processors and memory systems (*Stone et al., 2007*).

The parallel nature of ReMDFF presents an opportunity for efficient, automated sampling of maps of varying resolution. However, to achieve the best efficiency, the simulations should be performed on multi-core CPUs with relatively high core counts (i.e. at least 1 core per replica). Access to such multi-core computers could prohibit use of ReMDFF, however access to machines with the necessary hardware is easily achieved through cloud computing. The cloud computing model provides researchers with access to powerful computational equipment that would otherwise be too costly to procure, maintain, and administer on their own. A particular obstacle is that structural modeling often involves the use of different software suites, such as VMD (*Humphrey et al., 1996*) for simulation preparation and Situs (*Wriggers, 2010*) for initial rigid-body docking or VMD, NAMD (*Phillips et al., 2005*), and Rosetta (*Leaver-Fay et al., 2011*) for iterative refinement of models (*Lindert and McCammon, 2015*). Cloud platforms can easily bundle different software packages used in a modeling workflow to guarantee their availablity and interoperability on a standardized system. Through the cloud version of our MDFF program suite a user does not need to be aware of any of the above mentioned technical issues.

To prove the feasibility of performing ReMDFF simulations on a cloud platform, we ran ReMDFF for the test system, carbon monoxide dehydrogenase (PDB 1OAO:chain C), on the Amazon Web Services (AWS) Elastic Compute Cloud (EC2) platform. For the purposes of testing ReMDFF on EC2, we ran benchmarks on a variety of compute-optimized EC2 instance types. The details of the instance types can be found in *Table 3*. We used the same 6 smoothed density maps as the previously discussed cMDFF and ReMDFF simulations in the Proof of Principle and, therefore, also 6 replicas. The files and information necessary to run ReMDFF on the test system using EC2 cloud computing resources are available at (http://www.ks.uiuc.edu/Research/cloud/).

## Acknowledgements

This work is supported by grants NIH 9P41GM104601, NIH 5R01GM098243-02, and NIH U54GM087519 from the National Institutes of Health. The authors gratefully acknowledge computer time provided by the NSF-funded Extreme Science and Engineering Discovery Environment (XSEDE) MCA93S028, Sriram Subramanian for extremely insightful discussions on the high-resolution cryo-EM methods in general, and the β-galactosidase study in particular, and Benjamin Barad from the Fraser group at UCSF for guidance on the use of the EMRinger program. The authors also acknowledge the Beckman Postdoctoral Fellowship program for supporting A Singharoy.

## Additional information

### Funding

| Funder | Grant reference number | Author |
| --- | --- | --- |
| National Institutes of Health | 9P41GM104601 | Abhishek Singharoy<br>Ivan Teo<br>Ryan McGreevy<br>John E Stone<br>Klaus Schulten |

| | | |
|---|---|---|
| National Science Foundation | MCA93S028 | Abhishek Singharoy<br>Ivan Teo<br>Ryan McGreevy<br>John E Stone<br>Klaus Schulten |
| Beckman Institute for Advanced Science and Technology, University of Illinois, Urbana-Champaign | | Abhishek Singharoy |
| National Institutes of Health | 5R01GM098243-02 | Ryan McGreevy<br>John E Stone<br>Klaus Schulten |
| National Institutes of Health | U54GM087519 | Klaus Schulten |

The funders had no role in study design, data collection and interpretation, or the decision to submit the work for publication.

### Author contributions

AS, IT, RM, Conception and design, Acquisition of data, Analysis and interpretation of data, Drafting or revising the article; JES, Conception and design, Drafting or revising the article; JZ, Analysis and interpretation of data, Drafting or revising the article; KS, Computational simulation technologies, Conception and design, Analysis and interpretation of data, Drafting or revising the article

### Author ORCIDs

Klaus Schulten, ⓘ http://orcid.org/0000-0001-7192-9632

# Additional files

### Supplementary files

• Supplementary file 1. Supplementary tables. (A) TRPV1 MDFF Results. (B) Structure quality indicators for TRPV1 structures. (C) MDFF for the TRPV1 TM region. (D) Structure quality indicators for γ-secretase. (E) Measures of fit for MDFF refinements of β-galactosidase prepared initially at 1000 K. (F) Structural quality indicators for MDFF-refined β-galactosidase prepared initially at 1000 K.

• Supplementary file 2. Initial and refined structures.

### Major datasets

The following previously published datasets were used:

| Author(s) | Year | Dataset title | Dataset URL | Database, license, and accessibility information |
|---|---|---|---|---|
| Darnault C, Volbeda A, Kim EJ, Legrand P, Vernede X, Lindahl PA, Fontecilla-Camps JC | 2003 | Ni-Zn-[Fe4-S4] and Ni-Ni-[Fe4-S4] Clusters in Closed and Open Alpha Subunits of Acetyl-Coa Synthase/Carbon Monoxide Dehydrogenase | http://www.rcsb.org/pdb/explore/explore.do?structureId=1OAO | Publicly available at the Protein Data Bank (accession no: 1OAO) |
| Liao M, Cao E, Julius D, Cheng Y | 2013 | Structure of TRPV1 ion channel determined by single particle electron cryo-microscopy | http://www.rcsb.org/pdb/explore/explore.do?structureId=3J5P | Publicly available at the Protein Data Bank (accession no: 3J5P) |
| Liao M, Cao E, Julius D, Cheng Y | 2015 | Structure of the capsaicin receptor, TRPV1, determined by single particle electron cryo-microscopy | http://emsearch.rutgers.edu/atlas/5778_summary.html | Publicly available at the EMDataBank (accession no: EMD-5778) |
| Bartesaghi A, Merk A, Banerjee S, Matthies D, Wu X, Milne J, Subramaniam S | 2015 | 2.2 A resolution cryo-EM structure of beta-galactosidase in complex with a cell-permeant inhibitor | http://www.rcsb.org/pdb/explore/explore.do?structureId=5A1A | Publicly available at the Protein Data Bank (accession no: 5A1A) |

| | | | | |
|---|---|---|---|---|
| Bartesaghi A, Merk A, Banerjee S, Matthies D, Wu X, Milne JL, Subramaniam S | 2015 | 2.2 A resolution cryo-EM structure of beta-galactosidase in complex with a cell-permeant inhibitor | http://emsearch.rutgers.edu/atlas/2984_summary.html | Publicly available at the EMDataBank (accession no: EMD-2984) |
| Bai X, Yan C, Yang G, Lu P, Ma D, Sun L, Zhou R, Scheres SHW, Shi Y | 2015 | Cryo-EM structure of the human gamma-secretase complex at 3.4 angstrom resolution. | http://www.rcsb.org/pdb/explore/explore.do?structureId=5A63 | Publicly available at the Protein Data Bank (accession no: 5A63) |
| Bai XC, Yan CY, Yang GH, Lu PL, Ma D, Sun LF, Zhou R, Scheres SHW, Shi YG | 2015 | Cryo-EM structure of the human gamma-secretase complex at 3.4 angstrom resolution | http://emsearch.rutgers.edu/atlas/3061_summary.html | Publicly available at the EMDataBank (accession no: EMD-3061) |
| Lu P, Bai XC, Ma D, Xie T, Yan C, Sun L, Yang G, Zhao Y, Zhou R, Scheres SHW, Shi Y | 2014 | Structure of a extracellular domain | http://www.rcsb.org/pdb/explore/explore.do?structureId=4UPC | Publicly available at the Protein Data Bank (accession no: 4UPC) |
| Lu PL, Bai XC, Ma D, Xie T, Yan CY, Sun LF, Yang GH, Zhao YY, Zhou R, Scheres SHW, Shi YG | 2015 | Three-dimensional structure of human gamma-secretase at 4.5 angstrom resolution | http://emsearch.rutgers.edu/atlas/2677_summary.html | Publicly available at the EMDataBank (accession no: EMD-2677) |
| Li X, Mooney P, Zheng S, Booth C, Braunfeld MB, Gubbens S, Agard DA, Cheng Y | 2015 | Thermoplasma acidophilum 20S proteasome | http://www.rcsb.org/pdb/explore/explore.do?structureId=3J9I | Publicly available at the Protein Data Bank (accession no: 3J9I) |
| Li X, Mooney P, Zheng S, Booth C, Braunfeld MB, Gubbens S, Agard DA, Cheng Y | 2015 | 3D reconstruction of archaeal 20S proteasome | http://emsearch.rutgers.edu/atlas/5623_summary.html | Publicly available at the EMDataBank (accession no: EMD-5623) |
| Bartesaghi A, Matthies D, Banerjee S, Merk A, Subramaniam S | 2014 | Structure of beta-galactosidase at 3.2-A resolution obtained by cryo-electron microscopy | http://www.rcsb.org/pdb/explore/explore.do?structureId=3J7H | Publicly available at the Protein Data Bank (accession no: 3J7H) |
| Bartesaghi A, Matthies D, Banerjee S, Merk A, Subramaniam S | 2015 | Structure of beta-galactosidase at 3.2-A resolution obtained by cryo-electron microscopy | http://emsearch.rutgers.edu/atlas/5995_summary.html | Publicly available at the EMDataBank (accession no: EMD-5995) |

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

**Appendix 1**

# 1 Theory of cascade MDFF

The MDFF potential is given by

$$U_{\mathrm{EM}}(\mathbf{R}) = \sum_i w_i V_{\mathrm{EM}}(\mathbf{r}_i), \tag{7}$$

where $\mathbf{R}$ is the system configuration, $i$ labels the atoms in the structure, $w_i$ and $\mathbf{r}_i$ are weight and position, respectively, assigned to atom $i$, and $V_{\mathrm{EM}}$ is the potential map obtained from the EM density map.

Subjecting the potential map $V_{\mathrm{EM}}$ to a Gaussian blur increases its widths in a manner described as follows. The Gaussian blur process produces a potential map $V_\sigma$ from the potential map $V_{EM}$ through convolution with a normalized Gaussian of specified width $\sigma$:

$$V_\sigma(\mathbf{r}) = \int d\mathbf{r}' \, G(\mathbf{r}; \mathbf{r}', \sigma) V_{\mathrm{EM}}(\mathbf{r}'), \tag{8}$$

$$G(\mathbf{r}; \mathbf{r}', \sigma) = A(\sigma) \exp\left(-\frac{||\mathbf{r} - \mathbf{r}'||^2}{2\sigma^2}\right), \tag{9}$$

$$A(\sigma) = \frac{1}{(2\pi\sigma^2)^{3/2}}. \tag{10}$$

One can characterize the resolution of $V_{\mathrm{EM}}(\mathbf{r})$ explicitly by assuming that it can be written as a sum of Gaussians.

$$V_{\mathrm{EM}}(\mathbf{r}) = \sum_n c_n G(\mathbf{r}; \mathbf{r}'_n, \sigma'_n), \tag{11}$$

where $c_n$ are weighting factors, $\mathbf{r}'_n$ and $\sigma'_n$ are, respectively, the centers and widths of the component Gaussians. Substituting the above expression into **Equation 8** yields

$$V_\sigma(\mathbf{r}) = \int d\mathbf{r}' \, G(\mathbf{r}; \mathbf{r}', \sigma) \sum_n c_n G(\mathbf{r}'; \mathbf{r}'_n, \sigma'_n) \tag{12}$$

$$= \sum_n c_n A(\sigma) A'_n(\sigma'_n) \int d\mathbf{r}' \exp\left(-\frac{||\mathbf{r} - \mathbf{r}'||^2}{2\sigma^2} - \frac{||\mathbf{r}' - \mathbf{r}'_n||^2}{2\sigma'^2_n}\right), \tag{13}$$

Evaluation of the above expression gives

$$V_\sigma(\mathbf{r}) = \sum_n C_n \exp\left(-\frac{||\mathbf{r} - \mathbf{r}'_n||^2}{2(\sigma^2 + \sigma'^2_n)}\right), \tag{14}$$

$$C_n = c_n A(\sigma) A'_n(\sigma'_n)(2\pi)^{3/2} \left(\frac{\sigma\sigma'_n}{\sqrt{\sigma^2 + \sigma'^2_n}}\right)^3 \tag{15}$$

$$= c_n \left[2\pi(\sigma^2 + \sigma'^2_n)\right]^{-3/2}. \tag{16}$$

Note that setting $\sigma = 0$ A in (**Equations 14–16**) recovers the expression for the initial map in **Equation 11**, i.e. $V_0(\mathbf{r}) = V(\mathbf{r})$. On the other hand, increasing $\sigma$ results in an increase in the

widths of the component Gaussians, and leads in turn to an increase in the range of the MDFF force $\mathbf{F}_\sigma(\mathbf{r})$, where

$$\mathbf{F}_\sigma(\mathbf{r}) = -\nabla V_\sigma(\mathbf{r}) \tag{17}$$

$$= \sum_n \frac{C_n}{\sigma^2 + \sigma'^2_n} \exp\left(-\frac{||\mathbf{r} - \mathbf{r}'_n||^2}{2(\sigma^2 + \sigma'^2_n)}\right)(\mathbf{r} - \mathbf{r}'_n). \tag{18}$$

## 2 MD simulation setup

Unless otherwise stated, all simulations reported in the present study used the following MD parameters. MDFF simulations were run in vacuum, maintained at 300 K via a Langevin thermostat and employed a time step of 1 fs. Secondary structure restraints were imposed using NAMD's Extra Bonds function to prevent loss of secondary structure due to strong MDFF guiding forces.

MD simulations were prepared using CHARMM-GUI (**Jo et al., 2008**) and run under NPT conditions, maintained at 303.15 K temperature and 1 atm pressure using a Langevin thermostat and barostat. All systems were parametrized using the CHARMM36 force field (**Klauda et al., 2010**; **Best et al., 2012**). Structures were solvated in explicit water (TIP3P model) boxes, with at least 15 Å separation between structure and water box boundaries. Particle-mesh Ewald electrostatics was used and the time step was 2 fs. For simulations of the reported structures of β-galactosidase, backbone atoms were held fixed while a minimization over 1000 time steps was performed, followed by 32-ns and 40-ns equilibration for the 3.2-Å and 2.2-Å structures, respectively. Following the equilibration step, production runs of 30 ns were performed for both structures.

In the case of TRPV1, the channel was embedded in a membrane of standard lipid composition POPE, POPC, POPG at ratio 2:1:1. Initial runs involved minimization over 1000 steps and 1-ns equilibration of the lipid tails, with all other atoms fixed. In the following MD run, minimization was performed over 5000 steps and equilibration was performed for 3 ns, with protein backbone atoms held fixed. Finally, the entire system was equilibrated for a further 6.4 ns. During the equilibration, C-terminal residues 752 to 762 were harmonically restrained because a substantial C-terminal segment was missing in the structure.

## 3 Preparation of initial test structures

The initial test structure of β-galactosidase was obtained by subjecting the reported structure (**Bartesaghi et al., 2014**) to a 4-ns MD simulation at a temperature of 300 K in vacuum and using secondary structure restraints. Trajectory frames recorded at 2-ps intervals were evaluated for backbone RMSDs with respect to the reported structure. A frame with an RMSD value of 7.6 Å (**Figure 2—figure supplement 2a**) and lowest global cross-correlation with respect to the reported map was picked to be the initial test structure. The structural quality measure of this model is provided in **Table 2**. A second search model was prepared by repeating the same protocol but at 1000 K. This search model is also characterized by an RMSD of 7.6 Å but now features a more distorted local structure as measured in terms of increased rotamer and Ramachandran outliers (**Supplementary file 1F**).

The initial test structure of TRPV1 was also derived from a reported structure (**Liao et al., 2013**). In order to render the disjointed reported structure contiguous for correct structural dynamics during simulation, the missing loop region (residues 503 to 506) was added by hand. Additionally, the substantial ankyrin repeat region (residues 111 to 198) was removed because the corresponding density was missing from the map. For the purpose of testing the robustness of cMDFF and contrasting its performance with that of direct MDFF, the structure

was distorted (see *Figure 2—figure supplement 2b*) during an interactive MD (*Stone et al., 2001*; *Grayson et al., 2003*) simulation, subjecting residues 199 to 430 in one subunit's extramembrane domain to a series of transformations, consisting roughly of a polar angle change of 15° toward the cytoplasmic pole followed by an azimuthal rotation of 30°, so that the backbone RMSD of the transformed region was about 22 Å relative to the original structure.

## 4 Protocol efficiency

In the present study, the convergence of fitting results of TRPV1 and β-galactosidase for different protocols can be evaluated through RMSD value employing the reported de novo structures as reference. *Figure 2—figure supplement 3* shows the time evolution of RMSD for cMDFF, ReMDFF, and direct MDFF for the two proteins, It should be noted that the plots do not include the final refinement step, which is the same across all three protocols.

cMDFF and ReMDFF can be seen to reach similar RMSD levels, outperforming direct MDFF. ReMDFF converges more quickly than cMDFF in either example. Of the 6 replicas employed for the ReMDFF of TRPV1 two resulted in poorly fitted structures, having become trapped in density minima even after exchanging with the lowest-resolution map chosen with $\sigma$ = 5 Å. All replicas can be monitored during the simulation and poorly fitted ones can be discarded by a user. It is also worth noting that the region of the TRPV1 map to which both cMDFF and ReMDFF successfully fitted the model, characterizes a diverse range of local resolutions from 4 Å to 6 Å challenging thus the conformational sampling capability of any flexible fitting technique. For the same reason, this region was avoided during Rosetta refinements of TRPV1 (*DiMaio et al., 2015*) but is addressed now via MDFF.

## 5 Global vs. local measures of fit

The cross-correlation coefficient calculated over an entire structure, termed global cross-correlation coefficient (GCC), has been a popular indicator of goodness-of-fit of a structure to a corresponding EM density map. However, averaging over the entire structure smears out potentially useful local structure information and in some cases, can be misleading (see *Figure 2—figure supplement 1*), since GCC cannot distinguish between correct and wrong assignments of residues to a given map region as long as the residues are equally well fitted.

Local measures of fit allow one to assess every part of the structure individually. For example, local cross-correlation coefficients (LCCs) (*Stone et al., 2014*) were tracked over the course of the cMDFF simulations of β-galactosidase and TRPV1 (*Figure 2—figure supplement 4*). The improvement in LCC of the majority of residues in each case lends greater confidence in the fitting result. At the same time, residues that have relatively lower LCCs can be identified and given attention (*Stone et al., 2014*; *McGreevy et al., 2016*).

## 6 Fourier shell coefficients

Fourier Shell Coefficients (FSCs) can be used as a means of evaluating quality of fit by comparing the degree of similarity between the original map and a simulated map derived from the structure to be evaluated, using the simulated map feature of VMD's MDFF package (*Trabuco et al., 2008*, *2009*) and the same voxel size as the original map.

In the present study, FSC curves for fitted TRPV1 and β-galactosidase structures were calculated via the FSC operation in SPIDER (*Shaikh et al., 2008*) using a shell width of 0.5 reciprocal space units, and resolution cutoff of half the voxel size. In the case of TRPV1, both the full structure and MDFF-fitted region were evaluated. The latter was obtained by applying

a mask of the region around residues 199 to 430 in the fitted structure to the simulated map, and in the reported structure to the original map.

As a means of summarizing comparisons by FSC, other studies have used 'integrated FSCs', a numerical measure obtained by integrating under the FSC plot within a predefined resolution interval. Two integrated FSC measures, corresponding to the intervals 3.4 Å to 10 Å and 5 Å to 10 Å, were obtained in the present study and tabulated in *Table 1*, and *Supplementary file 1A and 1C*.

## 7 Other simulations of β-galactosidase

Beyond the simulations reported in *Table 1*, further simulations were performed to explore the performance of MDFF within the contexts of further types of analyses. The first of these simulations was a direct MDFF simulation of the reported β-galactosidase structure, fitting only backbone atoms to the 3.2-Å map. The fitted structure was compared to the 'refined de novo' structure reported in Results. It was found that EMRinger scores were lower at 2.35 when only backbone atoms were fitted, compared to 4.23 when non-hydrogen atoms were fitted. This result suggests that even if the backbone is correctly placed, the MD force fields alone (i.e. CHARMM36 (*Best et al., 2012*) here) are incapable of providing sidechain geometries consistent with the map. Refinement of the sidechains will therefore require explicit fitting to the density, above and beyond the orientations captured by the force fields alone.

In the second simulation, the resulting structure of the cMDFF simulation was subjected to an equilibrium MD simulation in explicit solvent. As shown in *Figure 2—figure supplement 6*, the equilibrium RMSD fluctuations during this simulation ranged between 3.0 Å to 3.4 Å of the starting structure. It is worth noting that these RMSD values agree well with the 3.2 Å resolution limit of the β-galactosidase map. Thus, the result indicates that uncertainties of the map resolution reflects quantitatively the structural variations of the cMDFF-fitted β-galactosidase model at the room temperature. Consequently, this model is representative of the thermodynamic ensemble that the EM map characterizes. Further analysis of the quality of the 3.2-Å β-galactosidase model is presented in the Model Validation subsection of Results.

The third set of simulations takes advantage of a unique opportunity, presented by the availability of two different maps of the same structure, at resolutions of 3.2 Å and 2.2 Å, to compare the results of fitting β-galactosidase to maps of different resolutions. The reported structures were subjected to direct MDFF simulation for 0.7 and 1 ns for the 3.2-Å and 2.2-Å models, respectively. The RMSF for each residue is calculated over consecutive 10-ps windows during the fitting. The RMSF values for all residues, including those for the PETG binding pockets (*Bartesaghi et al., 2015*) are plotted in *Figure 4—figure supplement 4*, reflecting smaller fluctuations during the fitting to the 2.2-Å map than in the 3.2-Å one. The relationship between fluctuation and map quality is examined in greater detail in Results, and imply that the RMSF of the fitted structure correlates negatively with the resolution of the corresponding map. Thus, the smaller fluctuations of the 2.2 Å structure than those of the 3.2 Å one further validate our proposed RMSF-map quality relationship of *Figure 4*.

## 8 MDFF refinement for TRPV1

Similar to the β-galactosidase results in *Table 1*, the refinement of TRPV1 was significantly better with cMDFF and ReMDFF than with direct MDFF, but starting with much poorer search models than what were employed for the β-galactosidase refinements. The TRPV1 refinement results are summarized in *Supplementary file 1A*: (i) RMSD of the fitted structure with respect to the reported de novo structure is 7.9 Å for direct MDFF, higher than the 2.4 Å and 2.5 Å RMSD values for cMDFF and ReMDFF, respectively (*Figure 2—figure supplement 3b*); (ii)

EMRinger scores for cMDFF and ReMDFF are 1.68 and 1.99 respectively, higher than the 1.51 score obtained for direct MDFF; (**iii**) MolProbity scores (**Chen et al., 2010**) are 2.4 and 2.5 for cMDFF and ReMDFF, smaller than the 7.9 score for direct MDFF, implying fewer, less severe steric clashes and fewer poor rotamers in the former than in the latter; (**iv**) integrated FSC (iFSC2, for the range 3.4-10 Å obtained as described in Appendix 1 - Section 6 above), attains higher values of 2.62 and 2.75 for cMDFF and ReMDFF respectively, than the 1.79 value for direct MDFF. iFSC1, corresponding to the lower resolution range of 5-10 Å is found to behave similarly to iFSC2; and (**v**) GCCs improve from an initial value of 0.16 to 0.50, 0.54 and 0.53 for direct, cMDFF, and ReMDFF protocols, respectively. Similarly, typical residue LCC values improve from 0 to 0.5 or higher, as shown in **Figure 2—figure supplement 4b**.

Measures of structural quality for the above fits are tabulated in **Supplementary file 1B**. Cross-validation with half-maps was also performed on the cMDFF structure, as per the β-galactosidase simulations, to ensure that it was not overfitted. As in the case of β-galactosidase, iFSC and EMRinger scores for direct and cross comparisons were similar. FSC analysis results are described in Appendix 1 - Section 11.

## 9 MDFF fitting of TRPV1 TM domain

Noting that the transmembrane (TM) region of the TRPV1 map is better resolved than the map for the extramembrane region, we performed 200 ps of MDFF on a truncated TM portion of the structure (residues 381 to 695) to characterize the performance of MDFF in the transmembrane region alone.

The TM region had previously been refined employing Rosetta tools (**Leaver-Fay et al., 2011**), providing an opportunity for comparison. Two MDFF simulations were performed, the first with only non-hydrogen sidechain atoms coupled to the density and harmonic restraints holding backbone atoms in the configuration of the reported structure (**Liao et al., 2013**) and another with all non-hydrogen atoms coupled to the density and the backbone restraints removed.

MDFF characteristics for fitting the isolated TM region of TRPV1 are summarized in **Supplementary file 1C**. Quality of fit measures, namely EMRinger and iFSC, for the backbone-restrained simulations were lower than those of the Rosetta-derived structure. However, MolProbity scores for the MDFF-derived structures are better than those of Rosetta. Allowing the backbone to be fitted into the map without restraints from the reported structure substantially improved the quality of fit measures so that they are comparable to those of Rosetta's, while maintaining a lower MolProbity score.

## 10 RMSF as a local measure of map quality

Per-residue Root mean square fluctuation (RMSF) has been found to be a good local measure of model fit, decreasing in relation to the increasing quality of fit over a cMDFF simulation, as illustrated in **Figure 4—figure supplement 1**. In addition to being a measure of fit, RMSF values can also be used for the analysis of local map quality. When monitored during MDFF refinement simulations, RMSF values serve as a good measure of local map quality, decreasing linearly with increase in local map resolution, as demonstrated in a number of test cases (**Figure 4**), including the 20S proteasome (**Figure 4—figure supplement 3**). In contrast to RMSF, other local measures of fit, such as LCC and EMRinger scores, are not necessarily good indicators of local map quality, as is evident from their poor correlation with local map resolution in **Figure 4—figure supplement 2**.

## 11 Cross-validation of MDFF-fitted structures

To demonstrate that the over-fitting does not occur during MDFF refinements, which is also fairly representative of cMDFF and ReMDFF refinements, the reported de novo structures of β-galactosidase and TRPV1 were each directly fitted to two half-maps (labelled 1 and 2) of the corresponding reported EM map, (*Bartesaghi et al., 2014*) for β-galactosidase and (*Liao et al., 2013*) for TRPV1. Subsequently, simulated maps were created from the fitted structures using VMD's MDFF plugin and resolution settings equivalent to the reported maps. In total, there were two simulated maps, also with labels 1 and 2 corresponding to the half-map from which the fits were obtained, for each protein. FSC plots describing the direct comparison of simulated maps with the corresponding half-maps (e.g. simulated map 1 with half-map 1) as well as the cross comparison of simulated maps with the non-corresponding half-maps (e.g. simulated map 1 with half-map 2) were created. *Figure 2—figure supplement 8* show the plots for β-galactosidase and TRPV1, respectively. The high degree of similarity between the cross comparisons as well as between cross comparisons and direct comparisons indicate a very low degree of over-fitting. In fact, iFSC values calculated for the plots (see Figures) are practically uniform. EMRinger scores for the same sets of comparisons were also calculated. For β-galactosidase, the EMRinger scores were 3.25 for simulated map 1 against half-map 1, 2.97 for simulated map 2 against half-map 2, 2.92 for simulated map 1 against half-map 2, and 2.81 for simulated map 2 against half-map 1; these numbers are fairly comparable to the EMRinger scores with the full maps as presented in *Table 1*. For TRPV1 again, the EMRinger scores were 1.43 for all comparisons. The high degree of similarity between EMRinger scores for the different comparisons corroborates the favorable conclusion drawn from the FSC calculations.

