## [Decision Letter]

Thank you for submitting your article "Ensemble-based model refinement and validation with Resolution Exchange MDFF for sub-5 Å cryo-electron microscopy maps" for consideration by *eLife*. Your article has been favorably evaluated by John Kuriyan (Senior editor) and three reviewers, one of whom is a member of our Board of Reviewing Editors. The reviewers have opted to remain anonymous.

The reviewers have discussed the reviews with one another and the Reviewing Editor has drafted this decision to help you prepare a revised submission.

Summary:

The novelty of this paper lies in the combination of a variety of tools and concepts, none of which individually are new. Moreover, the authors published work on MDFF fitting of EM maps. However, this new combination (MD-based fitting of EM maps, combined with replica exchange and gradual resolution increase) works very well in the test cases that are presented in this work. The radius of convergence is impressive.

The main application of this combined method is for cases where the initial model is very far from the EM map. However, in practice the applications of their method may be somewhat limited since either (a) the EM map is near atomic resolution (3.5 A or better) allowing ab initio tracing, or (b) the map is lower resolution in which case higher resolution structures of fragments are generally needed in order to have confidence in the final model. The new method is probably most useful in the resolution range 4-7 A where ab initio fitting to the maps is not possible and when high-resolution structures are available of all components, but these components undergo large conformational changes.

Along similar lines, the advantages of cMDFF/ReMDFF over 'direct' MDFF are clearly demonstrated for high-resolution, although it would be helpful to know if there are also advantages of using cMDFF/ReMDFF over direct MDFF at moderate resolutions (lower than 4 A resolution).

Essential revisions:

1) A common practice in the EM field is to perform refinements against half maps to judge the degree over-fitting. Assuming that the authors have access to the EM data for one of the test cases, such over-fitting test should be performed – as shown in DiMaio, et al. (2013) Protein Sci, 22, 865-868 or Amunts, et al. (2014) Science 343, 1485-1489.

2) It was unclear how well the protocols work at the second step – ensuring an accurate model that sufficiently describes the density at the side chain level. In the absence of figures demonstrating the fit of side chains to density (an omission that should be corrected), one of the reviewers compared the provided model coordinates (source data to Table 1) with the deposited structure of β-galactosidase (PDB accession code 3J7H). This revealed substantial problems with the model. There are a large number of Ramachandran outliers (~10%); more than 1% of non-prolines are in cis-configurations (less than 0.05% are expected); the backbone of the region around the galactoside binding site is incorrect, and many of the side chains are not in their correct density. The authors themselves note 'even if the backbone is correctly placed, the MD force fields […] are incapable of providing sidechain geometries consistent with the map'. This may represent a considerable limitation to using cMDFF/ReMDFF to refine atomic models against sub 5 Å EM maps. It would be helpful if the authors could demonstrate that cMDFF/ReMDFF is capable of refining good starting models without disrupting backbone and side chain geometries. This would more accurately reflect the starting model for many of the potential users of cMDFF/ReMDFF. For example, the deposited model for β-galactosidase should be refined with cMDFF and ReMDFF and not just direct MDFF. The manuscript should provide guidance on how to preserve side chain geometries and the prevent cis-peptide bond formation during refinement. This would improve the utility of cMDFF/ReMDFF as a method of fitting a well-refined structure to classes of different conformational states that can range from high to mid resolution – a good test case might be CorA (Matthies et al. 2016. Cell, 164(4):747-756). Further statistics should be provided to demonstrate the quality of the model after cMDFF/ReMDFF. For example, root-mean-square deviations for bond lengths and bond angles and Ramachandran statistics, which are typically minimal requirements when reporting the quality of structural models. These should be added to Table 1.

3) The authors propose per-residue root mean square fluctuation (RMSF) values as a way of specifying the precision of atom positions. These are shown to correlate with local map quality, suggesting they can be used for model-to-map validation. In many ways RMSF values appear to be analogous to atomic B factors. It would be helpful to see a side-by-side comparison of a protein structure colored by local resolution, atomic B factors and RMSF (an expansion of Figure 3). One would expect them to correlate in a way that EMRinger and local correlation coefficient do not (Figure 4—figure supplement 2). What are the advantages of RMSF over B factors, which are a more established measure? Do the authors actually recommend depositing ensembles?

4) It is unclear what advantages are to be gained by utilizing RMSF values to determine the B factor for sharpening the map (which involves multiple simulations) compared to Guinier analysis performed during the standard post-processing procedure when both provide the same, or similar, values (Figure 5—figure supplement 2). Could RMSF values be calculated for different parts of the model to generate locally sharpened maps that subsequently improve refinement (although one suspects that applying a mask during post-processing would result in a similar effect)?

5) There is no relation of the presented method with "phase extension" approaches in crystallography where phase experimental phase information is generally much less accurate than amplitude information, and phase extension method make use of the inherently higher accuracy of the amplitude information along with approaches such as solvent flattening of map averaging. The cMDFF and ReMDFF approaches do not actually improve the EM map, they just use filtered less rugged density maps for better convergence of the global conformational changes. Please avoid this analogy.

6) The title is a bit misleading, reading "Ensemble-based refinement…". One might expect an ensemble refinement method, which this is not. Similar with the expression "ensemble-based flexible fitting" (end of Discussion): the flexible fitting itself is not "ensemble-based". Ensemble-based here simply refers to the calculation of RMSF value from the trajectory.

7) Figure 5: The plot describes map sharpening. Sharpening implies applying a negative B-factor, therefore one expects negative B-factor values on the x-axis.

8) This sentence – "well within the structural uncertainty represented by the 3 Å resolution limit of the crystallographic data" –, in the Results, is a bit misleading, since it sounds as if the structural uncertainty for 3 Angstroms data is 3 Å, which is not the case but unfortunately a common misconception. The accuracy of atomic positions is usually considerably higher than the limiting resolution of a crystal. Please remove or reword this sentence.

9) The authors should discuss in more detail the limitations and potential applications of their method as suggested in the summary above.

10) Overall, the manuscript should be edited to remove repetition (cloud computing is described as being cost-effective four times in the paper) and inaccuracies. The Materials and methods section is almost indistinguishable in style from the Results section and could be shortened considerably, with some of the more insightful descriptions moved to the Results (for example the first paragraph of the subsection “Fluctuation Analysis”). Additionally, many of the reference choices seem unnecessary. For example, in the first paragraph of the Introduction, describing how cryo-EM has evolved, the authors cite papers on NMR methods for complexes over 20 kDa (Clore and Gronenborn), membrane proteins by XFEL (Neutze et al.) and ribosome structures at mid-resolution (the papers by Rawat et al.). It would be better to cite one of the recent reviews on cryo EM, for example 'Cheng, Y. 2015. Single-particle cryo-EM at Crystallographic resolution. Cell, 161(3):450-7.

11) All starting and refined models should be provided as supplementary data files.

---

## [Author Response]

*Essential revisions:*

*1) A common practice in the EM field is to perform refinements against half maps to judge the degree over-fitting. Assuming that the authors have access to the EM data for one of the test cases, such over-fitting test should be performed – as shown in DiMaio, et al. (2013) Protein Sci, 22, 865-868 or Amunts, et al. (2014) Science 343, 1485-1489.*

We have performed half-map cross-validation analyses for our β-galactosidase and TRPV1 test systems, to demonstrate that little to no over-fitting has occurred. We report the details of these analyses in [Supplementary-material SD1-data]–section 11 and Figure 2—figure supplement 8.

*2) It was unclear how well the protocols work at the second step – ensuring an accurate model that sufficiently describes the density at the side chain level. In the absence of figures demonstrating the fit of side chains to density (an omission that should be corrected), one of the reviewers compared the provided model coordinates (source data to Table 1) with the deposited structure of β-galactosidase (PDB accession code 3J7H).This revealed substantial problems with the model. There are a large number of Ramachandran outliers (~10%); more than 1% of non-prolines are in cis-configurations (less than 0.05% are expected); the backbone of the region around the galactoside binding site is incorrect, and many of the side chains are not in their correct density. The authors themselves note 'even if the backbone is correctly placed, the MD force fields […] are incapable of providing sidechain geometries consistent with the map'. This may represent a considerable limitation to using cMDFF/ReMDFF to refine atomic models against sub 5 Å EM maps. It would be helpful if the authors could demonstrate that cMDFF/ReMDFF is capable of refining good starting models without disrupting backbone and side chain geometries. This would more accurately reflect the starting model for many of the potential users of cMDFF/ReMDFF. For example, the deposited model for β-galactosidase should be refined with cMDFF and ReMDFF and not just direct MDFF.*

We concur with the reviewer that the previously submitted refined structure of β-galactosidase suffered from unfavorable structural quality statistics. In fact, several of the model validation criteria, including the EMRinger scores that evaluate detailed side-chain conformations seem quite insensitive to the local structural discrepancies. Fortunately, a closer analysis of the MolProbity statistics revealed that the structural defects in our reported model of β-galactosidase were inherited from a poor choice of starting structure. We addressed this issue at the outset of this letter (see first key improvement).

As noted by the reviewer, starting structures in actual use cases tend to be more structurally sound than the one used in the original submission, in particular with regards to Ramachandran outliers. Consequently, we have performed a new set of simulations of β-galactosidase using a better starting structure, i.e., one with fewer outliers, while still maintaining an RMSD of >7 Å w.r.t. the target structure to demonstrate a high radius of convergence of cMDFF and ReMDFF (subsection “Refinement of β-galactosidase”). The resulting fitted structures exhibit favorable structural statistics, demonstrating little to no disruption of backbone and sidechain geometries. Figure 2—figure supplement 7 is newly added to show examples of proper sidechain fitting within the density map.

*The manuscript should provide guidance on how to preserve side chain geometries and the prevent cis-peptide bond formation during refinement.*

The new set of refinements (Table 1 and Table 2), together with the one starting with the poor initial search model ([Supplementary-material SD2-data]) and another one starting with the reported model (PDB: 3J7H, Table 1 (refined de novo)) allow a systematic analysis of the quality of MDFF out- put as a function of β-galactosidase starting models (subsection “Refinement of β-galactosidase”). This analysis provides a guideline for the choice of initial models with MDFF refinements (in the aforementioned subsection).

*This would improve the utility of cMDFF/ReMDFF as a method of fitting a well-refined structure to classes of different conformational states that can range from high to mid resolution – a good test case might be CorA (Matthies et al. 2016. Cell, 164(4):747-756).*

As noted above, cMDFF/ReMDFF refinements have now been repeated with the reported β- galactosidase (3J7H).

We believe that the set of test systems employed now adequately showcases the advances made in MDFF. TRPV1, in particular, represents a map featuring a diverse range of local resolutions (3.4 Å to ~ 7 Å), which challenges the model refinement capabilities of MDFF across high to mid resolutions.

Nonetheless, it is noted in the Discussion section that MDFF provides the perfect tool to model structures within maps that capture a conformational transition, as is the case with the Mg^2+^ transporter reported by Matthies et al. A study of conformational transitions involving high- resolution EM data requires a separate publication, and is thus avoided in the current manuscript.

*Further statistics should be provided to demonstrate the quality of the model after cMDFF/ReMDFF. For example, root-mean-square deviations for bond lengths and bond angles and Ramachandran statistics, which are typically minimal requirements when reporting the quality of structural models. These should be added to Table 1.*

Table 1 has been updated to reflect the new results, and two new tables, Table 2 and [Supplementary-material SD2-data], have been added to report the structural statistics, including RMS deviations for bond lengths and angles, and Ramachandran statistics for both β-galactosidase and TRPV1.

*3) The authors propose per-residue root mean square fluctuation (RMSF) values as a way of specifying the precision of atom positions. These are shown to correlate with local map quality, suggesting they can be used for model-to-map validation. In many ways RMSF values appear to be analogous to atomic B factors. It would be helpful to see a side-by-side comparison of a protein structure colored by local resolution, atomic B factors and RMSF (an expansion of Figure 3). One would expect them to correlate in a way that EMRinger and local correlation coefficient do not (Figure 4—figure supplement 2). What are the advantages of RMSF over B factors, which are a more established measure? Do the authors actually recommend depositing ensembles?*

Figure 3 is now updated to also include structures colored by the square of RMSF and local B- factors. The spatial variations in local-resolution, RMSF_2_ and atomic B-factors show excellent qualitative agreement. Furthermore, utilizing the well-established relationship between B-factors and RMSF, B=8π ^2^/3(RMSF_2_), B-factors can be computed from MDFF refinements. The MDFF derived B-factors are shown to be in good agreement with those reported for β-galactosidase (Figure 5—figure supplement 4). An additional advantage of monitoring RMSF during MDFF refinements is that it not only indicates the quality-of-map/model (subsection “RMSF and Quality of Map”), but also the quality-of-fit (subsection “RMSF and Quality of Map”). More advantages of computing RMSF are provided in the response to the next question.

It is indeed advisable to submit a representative ensemble of structures rather than a single one. The rationale for supporting an ensemble view is the following. A functional biomolecule is not restricted to a single structure. It rather represents a distribution of dynamical structures all of which contribute to the well-defined function. An EM map is a manifestation of this distribution of structures, more so because unlike crystallography, it captures the biomolecules in a native- like environment. Consequently, reporting a single structure does not suffice to elucidate the conformational diversity that the full distribution of functionally relevant structures manifest. We report an ensemble of 12 structures that represent the conformational diversity of β-galactosidase (Figure 5—source data 1). Even though the statistics of the contributing structures are comparable, the flexibility of the loop regions over the structured regions is clearly observed; such information is obscured if only a single model is submitted.

*4) It is unclear what advantages are to be gained by utilizing RMSF values to determine the B factor for sharpening the map (which involves multiple simulations) compared to Guinier analysis performed during the standard post-processing procedure when both provide the same, or similar, values (Figure 5—figure supplement 2).*

Overall B-factors of cryo-EM maps are typically calculated by Guinier analysis for maps with resolutions better than ~10 Å (Rosenthal and Henderson, 2003). Using a mask around specific regions of interest, estimation of local B-factors for different parts of a map is also possible. However, this method is limited to large domains of macromolecular complexes due to problems associated with tight masking of cryo-EM maps. For maps with resolutions better than ~3 Å, local B-factors could be estimated and refined in X-ray crystallography programs. However, most high-resolution cryo-EM maps have resolutions between ~3 to 5 Å. In this resolution range, B- factors could be estimated from MDFF and used as prior information to improve model building and refinement. For resolutions better than ~3 Å, MDFF-derived values may serve as initial estimates of B-factors to be further refined by crystallography programs. These points are now made in the last paragraph of the subsection “RMSF and per-residue B-factors”.

*Could RMSF values be calculated for different parts of the model to generate locally sharpened maps that subsequently improve refinement (although one suspects that applying a mask during post-processing would result in a similar effect)?*

Yes, domain-based sharpening is now demonstrated with RMSF computations for the soluble and transmembrane domains of TRPV1 (Figure 5—figure supplement 1).

*5) There is no relation of the presented method with "phase extension" approaches in crystallography where phase experimental phase information is generally much less accurate than amplitude information, and phase extension method make use of the inherently higher accuracy of the amplitude information along with approaches such as solvent flattening of map averaging. The cMDFF and ReMDFF approaches do not actually improve the EM map, they just use filtered less rugged density maps for better convergence of the global conformational changes. Please avoid this analogy.*

The analogy is now avoided.

*6) The title is a bit misleading, reading "Ensemble-based refinement…". One might expect an ensemble refinement method, which this is not. Similar with the expression "ensemble-based flexible fitting" (end of Discussion): the flexible fitting itself is not "ensemble-based". Ensemble-based here simply refers to the calculation of RMSF value from the trajectory.*

The title of the manuscript has been changed to “Molecular dynamics-based model refinement and validation for sub-5 Å cryo-electron microscopy maps”. The expression in the Discussion referred to by the reviewer no longer contains the term “ensemble-based”.

*7) Figure 5: The plot describes map sharpening. Sharpening implies applying a negative B-factor, therefore one expects negative B-factor values on the x-axis.*

We have updated Figure 5 and its Figure 5—figure supplement 1 and Figure 5—figure supplement 2 to reflect negative B-factors.

*8) This sentence – "well within the structural uncertainty represented by the 3 Å resolution limit of the crystallographic data" –, in the Results, is a bit misleading, since it sounds as if the structural uncertainty for 3 Angstroms data is 3 Å, which is not the case but unfortunately a common misconception. The accuracy of atomic positions is usually considerably higher than the limiting resolution of a crystal. Please remove or reword this sentence.*

We agree; the sentence is now removed.

*9) The authors should discuss in more detail the limitations and potential applications of their method as suggested in the summary above.*

Relevant points in terms of the choice of search model, B-factor determination and complementarity with de novomodel building methods have now been added in subsections “Refinement of β-galactosidase”, “RMSF and per-residue B-factors” and last paragraph of the Discussion respectively.

*10) Overall, the manuscript should be edited to remove repetition (cloud computing is described as being cost-effective four times in the paper) and inaccuracies. The Materials and methods section is almost indistinguishable in style from the Results section and could be shortened considerably, with some of the more insightful descriptions moved to the Results (for example the first paragraph of the subsection “Fluctuation Analysis”). Additionally, many of the reference choices seem unnecessary. For example, in the first paragraph of the Introduction, describing how cryo-EM has evolved, the authors cite papers on NMR methods for complexes over 20 kDa (Clore and Gronenborn), membrane proteins by XFEL (Neutze et al) and ribosome structures at mid-resolution (the papers by Rawat et al). It would be better to cite one of the recent reviews on cryo EM, for example 'Cheng, Y. 2015. Single-particle cryo-EM at Crystallographic resolution. Cell, 161(3):450-7.*

Repetition in the revised manuscript has been reduced and overlapping paragraphs in the methods with the results have now been removed entirely or moved into the results. Unnecessary references have been removed in favor of the suggested review article.

*11) All starting and refined models should be provided as supplementary data files.*

All models are provided with this resubmission.